# Click-based determination of accumulation of molecules in *Escherichia coli*

George M. Ongwae[1], Zichen Liu[1], Shasha Feng[2], Mahendra D. Chordia[1], Rachita Dash[1], Yuchen He[1], Mohammad Sharifian Gh[1], Brianna E. Dalesandro[1], Taijie Guo[3], Karl Barry Sharpless[4], Jiajia Dong[3], M. Sloan Siegrist[5,6], Wonpil Im[2] & Marcos M. Pires[1,7] ✉

Gram-negative bacterial pathogens pose a significant challenge in drug development because their outer membranes hinder the permeation of small molecules. The lack of widely adoptable methods for measuring the cytosolic accumulation of compounds in bacterial cells further hinders drug discovery efforts. To address this challenge, we report the development of the Chloroalkane Azide Membrane Permeability (CHAMP) assay, which we designed specifically to assess molecule accumulation in the cytosol of Gram-negative bacteria. The CHAMP analysis utilizes bioorthogonal epitopes anchored within HaloTag-expressing bacteria and measures the cytosolic arrival of azide-bearing test molecules through strain-promoted azide–alkyne cycloaddition. This workflow enables robust and rapid accumulation measurements of thousands of azide-tagged small molecules. Our approach consistently produces comprehensive accumulation profiles that surpass the scale of previous measurements in *Escherichia coli* (*E. coli*). We validated the CHAMP assay across various chemical and biological contexts, including hyperporinated cells, membrane-permeabilized cells, and *E. coli* strains with impaired TolC function, a key component of the efflux pump. The CHAMP platform provides a simple, high-throughput, and accessible method that enables the analysis of over 1000 molecules within hours. This technique addresses a critical gap in antimicrobial research and has the potential to accelerate the development of effective agents against Gram-negative pathogens.

The increased prevalence of multidrug-resistant Gram-negative bacterial infections is extremely alarming[1]. The CDC issued a report in 2019 that categorized bacterial pathogens based on their threat level. The list of pathogens considered as "urgent threats" was primarily populated by antibiotic-resistant Gram-negative bacteria and included *E. coli*[2]. Lack of treatment options will usher in the post-antibiotic era,

meaning that routine infections can become lethal and standard invasive medical procedures carry a much higher level of risk. Therefore, strategies that aim to address these threats have high significance. We pose, and there is wide agreement in the community[3–12], that the lack of robust and widely adoptable tools to measure the accumulation of molecules into bacteria has severely hampered

[1]Department of Chemistry, University of Virginia, Charlottesville, VA, USA. [2]Departments of Biological Sciences, Lehigh University, Bethlehem, PA, USA. [3]Institute of Translational Medicine, Zhangjiang Institute for Advanced Study, Shanghai Jiao Tong University, Shanghai, China. [4]Department of Chemistry, The Scripps Research Institute, La Jolla, CA, USA. [5]Molecular and Cellular Biology Graduate Program, University of Massachusetts, Amherst, MA, USA. [6]Department of Microbiology, University of Massachusetts, Amherst, MA, USA. [7]Department of Microbiology, Immunology, and Cancer University of Virginia, Charlottesville, VA, USA. ✉e-mail: mpires@virginia.edu

antibiotic drug discovery. The Golden Era of antibiotics leveraged naturally abundant small molecules that were readily identified using traditional methods; since the end of the era, this methodology has proven to be much more difficult to further mine for new antibiotics. The next phase of antibiotic drug discovery could potentially leverage the wealth of proteomics, genomics, and metabolomics data to design small-molecule agents that are potent and of high specificity. To accomplish this, the field fundamentally needs guiding principles describing the molecular determinants of small molecule permeation into Gram-negative bacterial cells, akin to Lipinski's rules of 5 (Ro5).

Gram-negative bacterial pathogens pose a significant challenge to antibiotic development due to their distinctive cell wall structure, particularly the outer membrane (OM), which restricts the penetration of small molecules into the periplasmic space (and then further into other subcellular compartments). It is currently understood that small molecules capable of traversing the OM either possess specific physicochemical properties that enable crossing of this barrier by passive diffusion or hijacking porins or other membrane-embedded proteins (Fig. 1a). Typically, minimum inhibitory concentration (MIC) analyses have been used as a proxy for drug accumulation. However, this

approach is problematic because MIC reflects antimicrobial activity, which is more indicative of target engagement, rather than directly measuring intracellular concentrations. It is therefore crucial to develop robust and widely adopted live cell assays that can independently measure the accumulation of small molecules across the OM, beyond just relying on MIC values. Developing such methods will provide more accurate insights into the accumulation behavior of potential molecules and ultimately aid in improving antibacterial drug development[13].

The field has consistently regarded liquid chromatography-tandem mass spectrometry (LC-MS/MS) as a premier technique for quantifying drug accumulation[14–17]. Indeed, devoid of the need for a chemical tag, this method presents significant advantages for measuring the uptake of molecules in bacteria. However, mass spectrometry, while advantageous for analyzing whole cell association, has principal limitations as currently reported: (a) restricted throughput capacity and (b) would ambiguously define molecule location unless additional, careful fractionation methods are included to show periplasmic or cytoplasmic accumulation[18], and (c) needs compounds to be mass active. This lack of subcellular localization is also a limitation

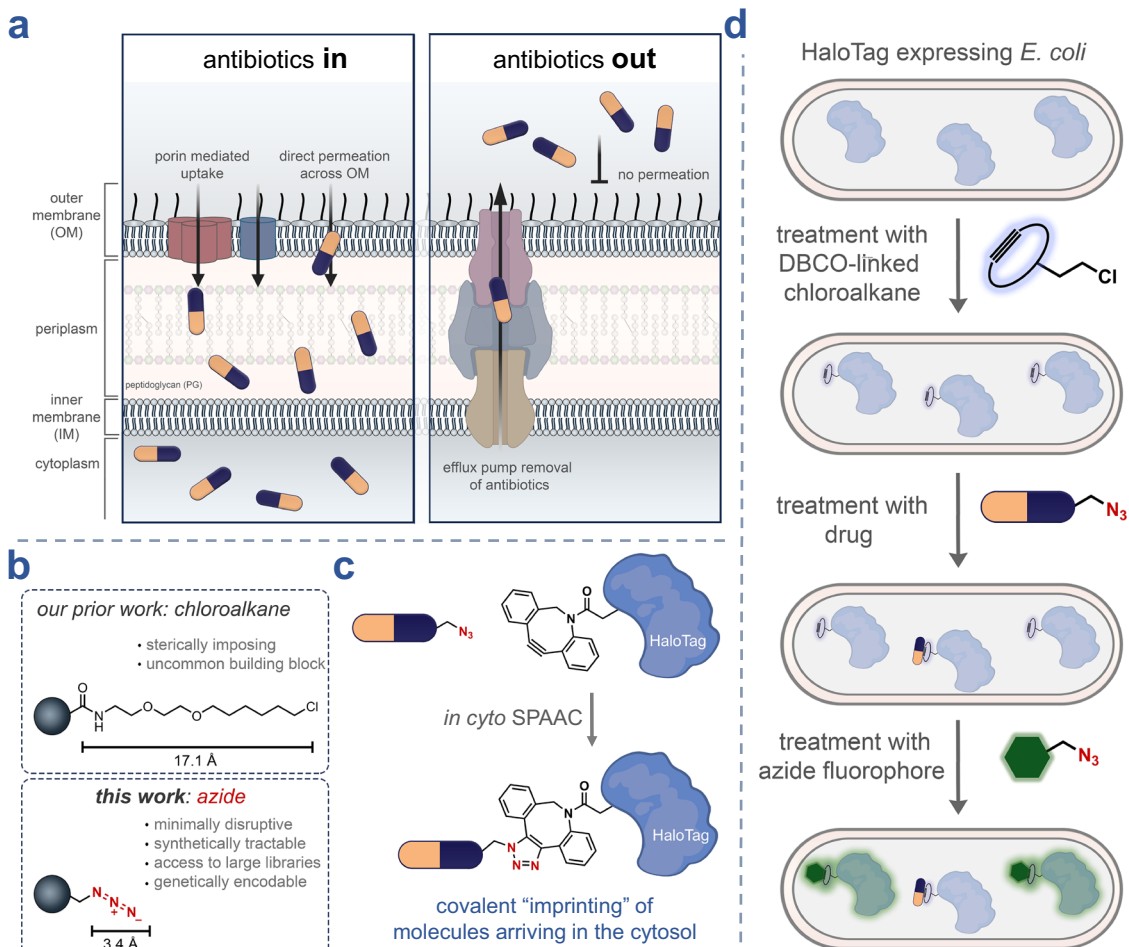

**Fig. 1 | Conceptual framework and rationale for the CHAMP assay. a** Schematic representation of the primary factors that can influence the level of molecules inside Gram-negative bacteria, including *E. coli*. There are several modes of permeation past the bilayers that surround Gram-negative bacteria, and these routes can differ based on the composition of the small molecule. **b** A direct comparison in molecular size between the chloroalkane tag that is compatible with HaloTag and the smaller azide tag. **c** Once compounds arrive in the cytosol of *E. coli*, they will be positioned to engage with the strained alkyne landmark that is found in the cytosol via a SPAAC reaction. This covalent register of arrival marks the subcellular localization of the drug within the cell. **d** Overall workflow of CHAMP in *E. coli*. First,

bacteria carrying a plasmid that expresses HaloTag in the cytosol are treated with a chloroalkane-modified strained alkyne that effectively converts the reactivity of HaloTag from chloroalkane to azide. The exposure of live cells to compounds of varying composition, which are azide tagged ('pulse step'), can be used to assess the level of cytosolic arrival. In the case of a molecule that reaches high levels of cytosolic arrival, there will be less strained alkyne to react with the subsequent exposure to a fluorescent azide ('chase step'). In the case of molecules with high levels of accumulation, the cellular fluorescence level is expected to be low. The opposite should be observed for molecules with low levels of cytosolic accumulation.

of spectroscopic measurements, such as those using fluorescent tags or inherent fluorescence. Despite these challenges, recent tour de force efforts by the Hergenrother laboratory[19,20] have successfully employed LC-MS/MS to develop the eNTRy rules, highlighting the significance of primary amines in the accumulation of small molecules in *E. coli*. Yet, this level of throughput has been difficult to achieve by others in the field. Consequently, limited throughput in analyzing the cytoplasmic accumulation of molecules remains a significant challenge. Without a clear understanding of the exact parameters that drive the accumulation of molecules into the cytosol of *E. coli*, efforts to broadly test and prioritize drug leads will be hindered.

In recent years, the Kritzer lab introduced a groundbreaking method for quantifying cytosolic accumulation of molecules in mammalian cells. This method, called the ChloroAlkane Penetration Assay (CAPA), involves applying a chloroalkane-tagged 'target molecule' to HaloTag-expressing mammalian cells[21,22]. The level of cytosolic accumulation is then inferred from the fluorescence signal produced by a chloroalkane-fluorophore 'chase molecule'. HaloTag, which can be expressed as freely diffusing proteins in the cytoplasm, selectively reacts with chloroalkanes to form a stable covalent bond. Critically, the covalent and subcellular nature of the assay confirms the cytosolic arrival of molecules, rather than whole-cell association[23–28]. For these reasons, this method is now widely regarded as the gold standard for measuring cytosolic accumulation in mammalian systems. Building on the success of CAPA in mammalian systems, we recently adapted this approach to measure cytosolic accumulation in diderm bacteria, such as *E. coli, Mycobacterium smegmatis, and Mycobacterium tuberculosis*, in a process we term 'Bacterial CAPA' (BaCAPA)[29]. To address (1) potential interference induced by the 15-atom chloroalkane tag in bacterial accumulation studies (Figs. 1b) and (2) limitations in the availability of chloroalkane-tagged test molecules, we developed an enhanced method in *E. coli* that substitutes the chloroalkane with a shorter 3-atom azide tag. The choice of the azide tag was based on its minimal size, relatively low impact on the physicochemical properties of the parent compound, and stability in biological systems[30,31]. Moreover, there are well-established protocols for late-stage azide installation into complex molecules[32,33] and in synthetic peptides, as well as metabolic incorporation within biomacromolecules, such as RNA, oligosaccharides, and proteins[34–37].

Our team recently introduced the Peptidoglycan Accessibility Click-Mediated Assessment (PAC-MAN) assay in mycobacteria[38]. In PAC-MAN, a strained alkyne, dibenzocyclooctyne (DBCO), is metabolically incorporated[39–41] into the peptidoglycan layer of diderm bacteria to monitor the passage of molecules through the OM. In this work, we adapted our previous workflow by treating HaloTag-expressing *E. coli* cells with a chloroalkane-DBCO reagent that anchors a strained alkyne to cytosolic HaloTag. Azide-tagged test compounds that reach the bacterial cytosol are then covalently imprinted to HaloTag via strain-promoted azide-alkyne cycloaddition (SPAAC)[42] reactions (Fig. 1c). This step is followed by treatment with an azide-tagged fluorophore, after which cellular fluorescence is measured using a flow cytometer or a plate reader in 96- or 384-well plates. Fluorescence levels are inversely proportional to the extent of SPAAC reactions between DBCO and the test azides in the cytosol, providing a straightforward method to determine apparent cytoplasmic arrival (Fig. 1d). The irreversible covalent reaction between chloroalkane-DBCO and HaloTag anchors DBCO in the cytosol and offers a method for determining subcellular localization. This adaptation, termed the 'CHloroalkane Azide Membrane Penetration' (CHAMP) assay, focuses on measuring cytosolic accumulation rather than whole-cell association. It operates at a throughput level that, to our knowledge, has not been previously achieved on such a scale. We demonstrate the capability to make more than 5000 measurements of approximately 1500 unique molecules in various biological contexts.

## Results

We first optimized the assay parameters to maximize the dynamic range of the fluorescence signals and adapted the assay for compatibility with high-throughput screening platforms. A small panel of azide-tagged molecules was used to benchmark the testing phase, followed by testing a series of antibiotics with varying levels of biological activity against *E. coli*. Next, this approach was validated by evaluating the impact of OM permeabilization in *E. coli* on the accumulation of azide-modified drugs and analyzing the reactivity of these azides with DBCO-modified beads. Furthermore, the assay was adapted for high-throughput screening platforms, with the datasets coupled with structural modifications that were previously proposed to impact accumulation in *E. coli*. We analyzed a highly diverse set of azide-tagged molecules, including a 404-member and a 1152-member library, to evaluate their cytosolic uptake in *E. coli* under four unique biological contexts: wildtype, *tolC*-null *E. coli* (the major efflux pump gene), porin overexpression (highly permeable/hyperporinated), and OM permeabilization induced by antibiotic treatment. These thousands of unique measurements, all acquired in under a month, were analyzed to deliver a first-in-class profiling platform in *E. coli*.

### Optimization of HaloTag Expression in *E. coli*

Optimizing key conditions in the CHAMP assay is essential before using it as a method for assessing molecule accumulation in bacterial cells. These conditions include HaloTag expression, anchoring of DBCO-chloroalkane onto HaloTag, choice of fluorophore, and conditions for measuring the accumulation of the test molecules. First, *E. coli* cells were transformed with a plasmid encoding HaloTag (see SI for details). The expression of functional HaloTag was confirmed by treating cells with a chloroalkane-linked rhodamine 110 (R110cl) and subsequently conducting flow cytometry analysis (Fig. 2a). Given the nature of the dye, it was expected to have a high level of cellular accumulation. The observed cellular fluorescence intensity correlated well with the concentrations of IsoPropyl β-D-1-ThioGalactopyranoside (IPTG) used during the protein expression phase (Fig. 2b). Of significance, the cellular fluorescence intensities initiated at a low value in the absence of IPTG, suggesting that the expression of HaloTag was responsible for capturing the chloroalkane-linked fluorophore in the cytosolic space of *E. coli*. The signal intensities exhibited exponential growth with increasing IPTG concentration before reaching a saturation plateau.

Additionally, with the goal of optimizing the concentration of the anchoring chloroalkane ligands in the CHAMP assay, cells expressing high levels of HaloTag were subjected to varying concentrations of R110cl. Cellular fluorescence intensities were low in the absence of R110cl, while they experienced significant enhancement at 0.3 μM of R110cl, followed by a dose-dependent increase that plateaued at approximately 3 μM (Fig. 2c). The localization of the fluorescence signals within the cells was additionally assessed using confocal imaging (Figs. 2d and S1). To confirm that the accumulation measurements in this study reflect cytoplasmic localization of molecules, the cytoplasmic compartment of *E. coli* cells was delineated by labeling the cell wall with D-Lys(FITC)-OH, which labels the peptidoglycan layer[43]. Consequently, the cytosolic probe for confocal imaging was changed to coumarin chloroalkane (Comcl), which fluoresces in a distinct spectral window, thereby enabling clear spatial separation of cytosolic and periplasmic signals. Moreover, fluorescence imaging of an SDS-PAGE gel containing cellular components from induced cells revealed a fluorescent band at approximately 30 kDa, consistent with the expected molecular weight of HaloTag (Fig. 2e). Together, these results confirm the capability of HaloTag to selectively anchor molecules tagged with a chloroalkane linker in *E. coli*.

## DBCO-chloroalkane anchoring within HaloTag

To mitigate the potential impact of a longer chloroalkane linker, we employed an azide tag that could then undergo an *in cyto* SPAAC

reaction[30,31] with DBCO. To accomplish this, we envisioned that we could anchor a DBCO epitope within HaloTag by pre-treating cells with DBCOcl (Fig. 2f). HaloTag-bound DBCO can then be leveraged to

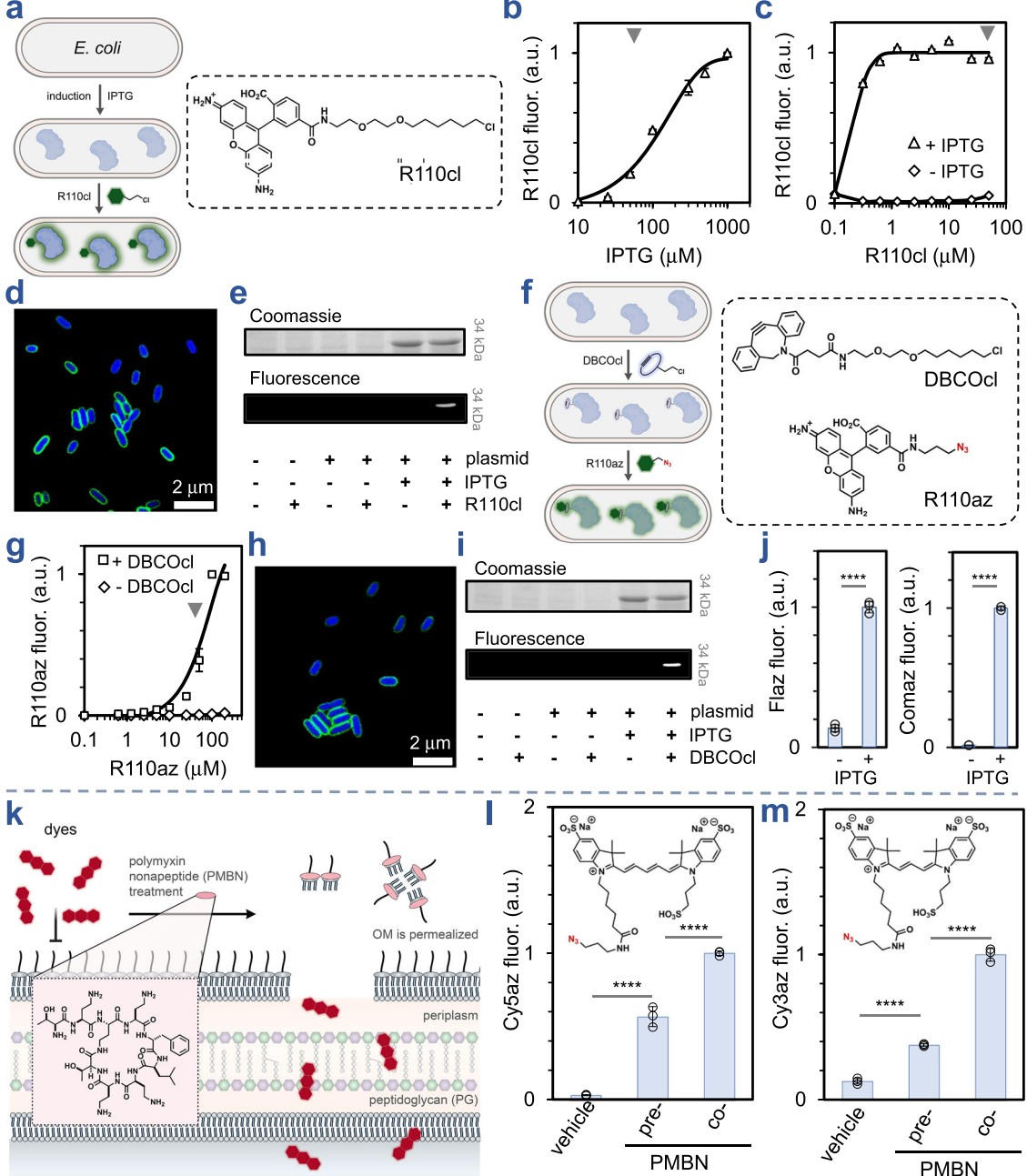

**Fig. 2 | Development and validation of the CHAMP assay. a** Scheme of the analysis of HaloTag expression in *E. coli*. HaloTag-expressing cells were incubated with R110cl, which should react with HaloTag in the cytosol. The chemical structure of R110cl is shown on the right. **b** Cellular assay with *E. coli* carrying the HaloTag-expressing plasmid titrated with varying levels of IPTG. **c** Cellular assay with HaloTag-expressing *E. coli* titrated with the ligand R110cl. **d** Fluorescence confocal images of D-Lys(FITC) labeled *E. coli* expressing HaloTag and treated with Comcl; this. observation was reproducible **e** SDS-PAGE analysis (Coomassie and fluorescence) of *E. coli* carrying the HaloTag-expressing plasmid or the empty vector in the presence/absence of either IPTG or R110cl. **f** Scheme of the analysis of HaloTag expression in *E. coli* with the CHAMP format. HaloTag-expressing cells were incubated with DBCOcl, which should react with HaloTag in the cytosol, followed by treatment with R110az. The chemical structures of DBCOcl and R110az are shown on the right. **g** Cellular assay with HaloTag-expressing *E. coli* after treatment with DBCOcl, then titrated the fluorophore R110az. **h** Fluorescence confocal images of

D-Lys(FITC) labeled *E. coli* expressing HaloTag, treated with DBCOcl, then reacted with Comaz. **i** SDS-PAGE analysis (Coomassie and fluorescence) of *E. coli* carrying the HaloTag-expressing plasmid in the presence/absence of IPTG, DBCOcl, and treated with R110az. **j** Cellular assay with HaloTag-expressing *E. coli* after treatment with DBCOcl then treatment with the fluorophore Flaz (left) and Comaz (right). **k** Schematic representation of how PMBN can permeabilize the OM of *E. coli* and facilitate greater apparent accumulation of less permeable molecules. **l** Cellular assay with HaloTag-expressing *E. coli* after treatment with DBCOcl, then treated with the fluorophore Cy5az. Cells were treated with vehicle, pre-incubated with PMBN, or co-incubated with PMBN. **m** Cellular assay with HaloTag-expressing *E. coli* after treatment with DBCOcl, then treated with the fluorophore Cy3az. Cells were treated with vehicle, pre-incubated with PMBN, or co-incubated with PMBN. Data are represented as mean +/− SD (*n* = 3) of technical replicates. *P*-values were determined by a two-tailed *t*-test (* denotes a *p*-value < 0.05, ** < 0.01, ***<0.001, ****<0.0001, ns = not significant).

assess the accumulation of azide-bearing target molecules in the cytosol of *E. coli*. Several experiments were conducted to refine the procedure. First, *E. coli* cells expressing HaloTag were incubated with DBCOcl to install the DBCO reporter within the cytoplasm. Empirical assessments were conducted to determine the optimal concentration for DBCOcl anchoring, employing an approach reminiscent of BaCAPA experiments. Cells were incubated with varying concentrations of DBCOcl followed by the treatment with R110cl (Fig. S2). Our findings indicated that the maximum levels of DBCO installation were achieved at approximately 50 μM of DBCOcl, exhibiting an $EC_{50}$ value of ~18.5 μM.

Subsequently, we conducted a baseline CHAMP experiment by treating *E. coli* cells with the azide-tagged rhodamine 110 (R110az) to reveal the anchored DBCO handles within individual cells. A titration of R110az revealed a concentration-dependent increase in fluorescence intensities, establishing the optimal concentration of R110az for subsequent experiments (Fig. 2g). Satisfyingly, a large dynamic range in fluorescence of up to 30-fold was observed in cells treated with both DBCOcl and R110az compared to controls. In the absence of DBCOcl treatment, cellular fluorescence levels were similar to those of untreated cells, indicating that R110az has minimal retention in bacterial cells without a covalent SPAAC reaction. Cellular fluorescence levels were also near background levels in non-induced cells that were also treated with DBCOcl and R110az, suggesting negligible retention of DBCOcl or cytosolic DBCO modified with R110az (Fig. S3).

Confocal microscopy confirmed that the fluorescence signal was distributed throughout the cytoplasm (Fig. 2h and S4) and fluorescent gel imaging also showed a band matching in the expected molecular weight range of HaloTag (Fig. 2i). These results are consistent with a site-selective anchoring of the DBCO and subsequent SPAAC with an azido-fluorophore at the site of HaloTag. When HaloTag was expressed in a smooth *E. coli* strain (ATCC 25922 with intact lipopolysaccharides on its outer surface), there was a similar dynamic range (Fig. S5). These data highlight the potential ability to translate CHAMP to a system with more complex surface composition and suggest that CHAMP can be applied more broadly across most types of bacteria that can be genetically manipulated. In the case of smooth *E. coli*, it is important to consider how the system can work in this context because surface biomacromolecules (including *O*-antigen) can alter accumulation profiles due to the potential impact of *O*-antigens in steric shielding of porins[44,45]. To verify the viability of the *E. coli* cells after inducing HaloTag expression and subjecting cells to DBCOcl treatment, we conducted a SYTOX Green analysis[46]. In live prokaryotic cells, SYTOX Green is excluded from cells with intact permeability barriers[47]. Our results revealed that the expression of HaloTag and incubation of these cells with DBCOcl and its occupancy by HaloTag in *E. coli* did not compromise the integrity of the cytoplasmic membrane (Figure S6). This finding emphasizes the suitability of utilizing CHAMP for investigating molecular permeation to the cytosol without causing disruption to the cell's permeability barriers.

We next sought to assess the compatibility of CHAMP with other azide-bearing fluorophores. Specifically, two more azide-bearing fluorophores, fluorescein (Flaz) and coumarin (Comaz), were tested. Incubation of cells after DBCOcl treatment with either fluorophore resulted in a significant increase in the cellular fluorescence (Fig. 2j). Cells incubated with Flaz exhibited a considerably lower fluorescence intensity compared to R110az-treated cells. Nevertheless, those results confirm the versatility of the assay for using various azide-bearing fluorophores. Recognizing that the assay is effectively completed at the end of the incubation with the test molecule, we proposed that higher signal-to-noise ratios could be achieved by removing the permeation barrier to the azido-fluorophores. Notably, formaldehyde fixation has been previously shown to outperform alcohol-based fixation in terms of increasing permeability in *E. coli*[48]. Our results indicated that fixation performed prior to the incubation with

fluorophores led to an overall increase in fluorescence levels for all three tested molecules (Fig. S7). This observation suggests that the dyes themselves could face significant permeation challenges imposed by the intact envelope of *E. coli*. Among the three dyes, the most favorable signal-to-noise ratio in fixed cells was achieved with Flaz treatment, resulting in a substantial increase in cellular fluorescence with minimum background staining.

## Influence of OM permeabilization on accumulation of Cy3az and Cy5az

We then applied CHAMP to analyze the impact of OM permeabilizers. Of significance, the permeation of small molecules in Gram-negative bacteria is purported to be primarily impeded by the OM[49–52]. This formidable barrier has been a focal point of previous investigations aimed to identify compounds capable of disrupting the OM integrity. These types of molecules can serve as antibiotic adjuvants to enhance the effectiveness of antibiotics or circumvent resistance mechanisms through co-treatment. A prominent example in this category is polymyxin B nonapeptide (PMBN)[53], a nonapeptide fragment derived from polymyxin B (Fig. 2k). PMBN is of particular interest due to its ability to permeabilize the OM at low concentrations. In our hands, PMBN displayed no significant impact on the colony-forming unit (CFU) count when administered at concentrations of 5 μM and 10 μM, consistent with previous studies demonstrating its low toxicity at these concentrations (Fig. S8)[54].

To analyze the impact of OM permeabilization by PMBN on the accumulation of molecules, we selected the sulfonated cyanine series as model compounds. These dyes contain an azide group, specifically Cy5az and Cy3az, whose permeabilities are limited by their sulfonate groups (Fig. 2l, m). We hypothesized that OM permeabilization by PMBN would enhance the accumulation of the dyes, thereby increasing cellular fluorescence. Briefly, live *E. coli* cells were either pre-exposed to low concentrations of PMBN or co-incubated with PMBN and dye to assess the accumulation of Cy5az and Cy3az. Our data revealed a striking 19-fold increase in cellular fluorescence when PMBN was pre-incubated with the cells, and an even more substantial 34-fold increase when the cells were co-incubated with PMBN and Cy5az (Fig. 2l). A full range of permeabilization by PMBN was observed by 2.5 μM with an $EC_{50}$ of 0.8 μM (Fig. S9). This pattern was mirrored with the analogous but smaller fluorophore, Cy3az, signifying that OM permeabilization assisted the accumulation of slightly less hydrophobic dyes as well (Fig. 2m). Given the substantial dynamic range that was observed, we propose that this assay could also be adapted for detection using a microplate reader for high-throughput screening. Indeed, our results showed that the dynamic range was mostly retained in this format (Fig. S10), which should provide a more direct path to screening campaigns with diverse molecular libraries for OM permeabilizers.

## Accumulation of test small molecules and antibiotics

We set out to establish the feasibility of CHAMP in monitoring the accumulation of azide-tagged molecules into *E. coli* cells. For this initial demonstration, *E. coli* cells with DBCOcl anchored on expressed HaloTag were titrated with a test azide-bearing molecule (Fig. 3a), followed by treatment with R110az, and cellular fluorescence levels were measured via flow cytometry. A concentration-dependent reduction in cellular fluorescence was observed, revealing an $EC_{50}$ of ~14 μM. The fluorescence intensity reached a maximum value at approximately 50 μM of the test molecule. These findings confirm the capability of the optimized CHAMP assay for investigating the accumulation of molecules in *E. coli*. With these results in hand, we shifted our focus to testing a series of azide-tagged small molecules featuring diverse structural motifs, anticipating that these variations would influence their accumulation. In the first subset, **1–3**, we tested the potential impact of amidation of carboxylic acids (Fig. 3b). Indeed, we

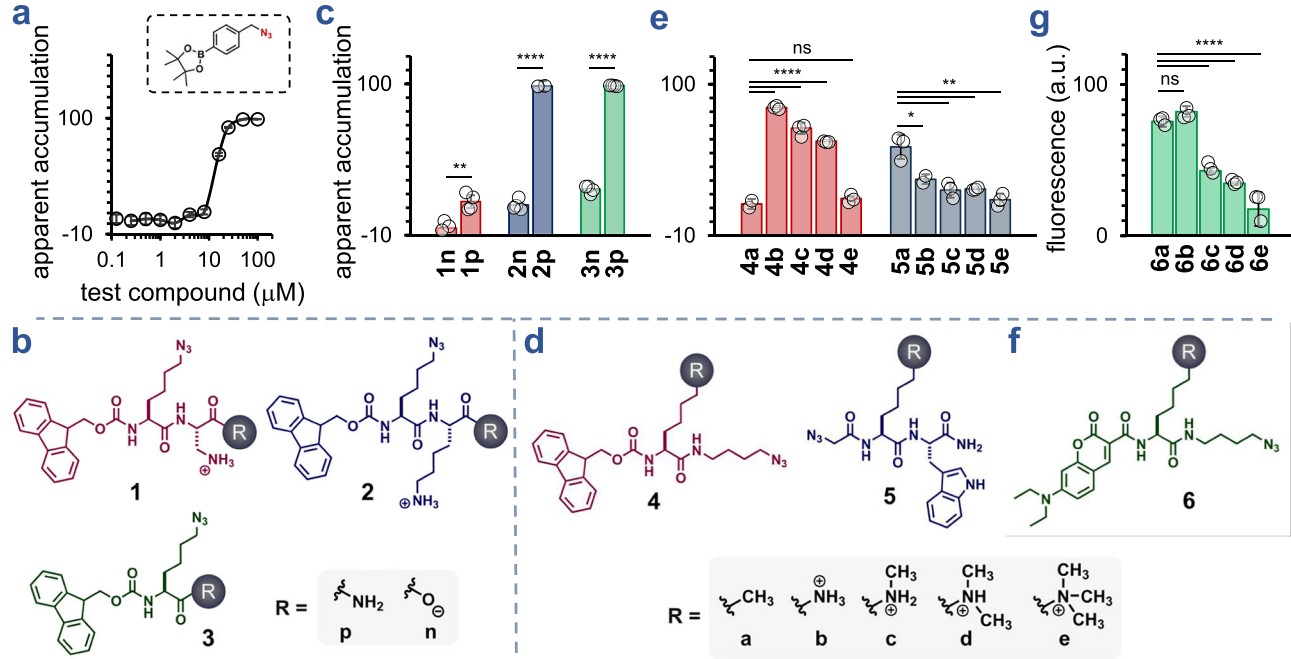

**Fig. 3 | Cytosolic accumulation of small-molecule test compounds. a** CHAMP analysis of *E. coli* titrated with an azide-tagged test molecule; the graph was obtained by plotting (1 − fold change, FC) against concentration, such that a higher (1 − FC) value corresponds to greater molecular accumulation. **b** Chemical structures of the **1**–**3** series. **c** CHAMP analysis of *E. coli* with 50 μM of **1n**, **1p**, **2n**, **2p**, **3n**, and **3p**. For all CHAMP assays, *E. coli* cells carrying the HaloTag-expressing plasmid were induced with IPTG, treated with DBCOcl, then incubated with the azide-tagged test molecules for 1 h in PBS at 37 °C. **d** Chemical structures of the **4**–**5** series. **e** CHAMP analysis of *E. coli* with 50 μM of series **4** and **5**. **f** Chemical structures of the series **6**. **g** Direct cellular fluorescence assay with HaloTag-expressing *E. coli* after treatment with DBCOcl then 50 μM of series **6**. Data are represented as mean +/− SD ($n = 3$ or 4) of technical replicates. *P*-values were determined by a two-tailed *t*-test (* denotes a *p*-value < 0.05, ** < 0.01, ***<0.001, ****<0.0001, ns = not significant).

observed that carboxamide, **1p**, had a higher level of apparent accumulation relative to its carboxylic acid counterpart **1n** (Fig. 3c). Appreciating the potential for an internal hydrogen bond due to the amino group in the β-position of *C*-terminal 2,4-diaminobutyric acid and the potentially confounding behavior of **1p/1n**, we extended the amino group further from the β-carbon in **2n/2p**. The apparent accumulation level of **2p** was significantly more pronounced, which may reflect the greater contribution of the positively charged lysine side chain, as it cannot readily form an internal hydrogen bond with the *C*-terminal group. While compounds **1n/1p/2n/2p** are theoretically expected to be primarily positively charged or net neutral under physiological pH, we sought to test if the pattern would hold true for a similar compound lacking a basic amino group. Our results showed that carboxamide in **3p** still led to higher levels of cytosolic accumulation in *E. coli* relative to the net negative **3n**. This small panel of compounds demonstrates how specific structural variations can be readily tested for their potential ability to drive accumulation in *E. coli*, which could be crucial in designing prodrugs that mask carboxylic acids to enhance their permeability in similar pathogenic bacteria.

The Hergenrother group has used an established method in the field, mass spectrometry, to empirically demonstrate that primary amines can serve as privileged functional groups to improve accumulation in *E. coli* as part of the eNTRy rules[17,19,20]. To benchmark *E. coli* CHAMP with these structural parameters, we made three additional compound panels, **4**–**6** (Fig. 3d). Series **4** has a similar general scaffold to our established molecules (**3**) that exhibited high levels of accumulation, while the molecules (**4a-4e**) feature five different configurations of the side chains ranging from an unfunctionalized terminal hydrocarbon to amino groups that vary in their degrees of methylation. Within this series, it was observed that the substitution of a methyl group in **4a** for a primary amino group in **4b** resulted in a large increase in apparent accumulation (Fig. 3e). The alkylation of the amino group progressively reduced accumulation levels in **4b-4e**.

These results show marked agreement with the eNTRy rules. In fact, the trimethyl **4e** had a similar accumulation profile as the compound **4a**, which does not have an amino group. To test the generalizability, a subsequent series, **5**, was synthesized that retained many of the same features to test the generalizability. Interestingly, the hydrocarbon-terminated **5a** displayed higher apparent accumulation than any other analog with an amino group, regardless of methylation states.

Next, we pivoted to benchmark the assay using an alternative strategy. Instead of using a pulse-chase modality to measure apparent accumulation, the test compounds in **6** all have a small fluorophore incorporated as part of the molecule (Fig. 3f). The goal was to determine whether the pulse-chase method produces results consistent with those obtained from a single-step direct fluorescence analysis. Similar to the **4** and **5** series, the methylation of the amino groups in series **6** led to lower accumulation levels (Fig. 3g). The absence of DBCOcl that installed the landmark and IPTG led to near background levels of fluorescence, which suggests that the cellular fluorescence observed in series **6** is due to engagement with the landmark and reported on cytosolic arrival (Fig. S11). Within the same series, we reasoned that we could evaluate the CHAMP-based levels of accumulation using a pulse-chase workflow. Consistently, we observed that the pattern of accumulation across **6a-6e** was nearly identical for a direct measurement and as a CHAMP-based analysis with a non-overlapping fluorophore (Fig. S12). Together, these findings align well with the eNTRy rules and underscore the advantages of having a robust and straightforward method for measuring cytosolic accumulation. Such a method can significantly enhance the throughput of identifying structural determinants of accumulation.

We posited that CHAMP would serve as a versatile platform for unraveling the molecular determinants of accumulation related to elements found within the OM, which can include both the membrane bilayer and the proteins embedded within this barrier. To achieve this, we sought to compare the apparent accumulation of azide-tagged

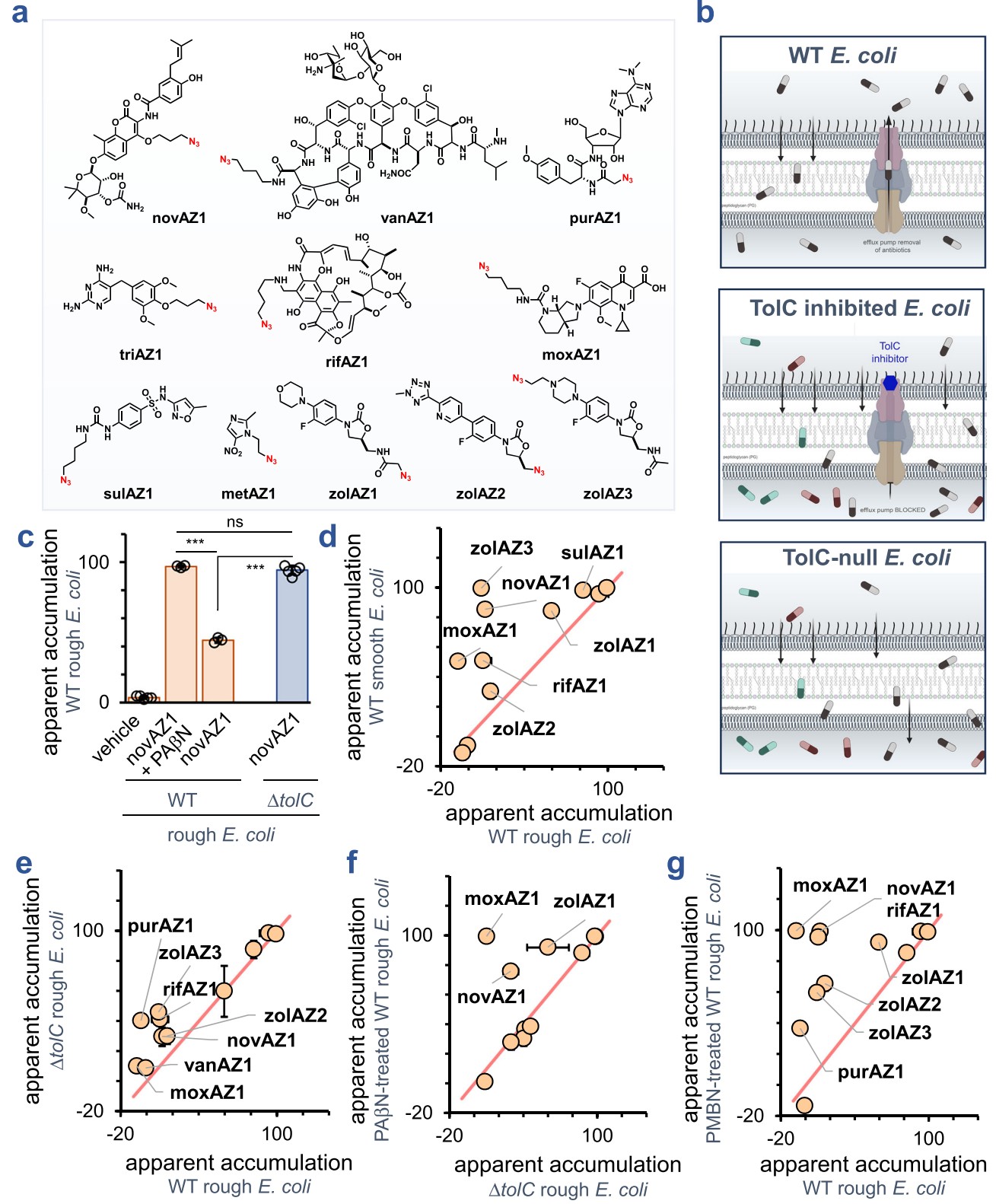

antibiotics (Fig. 4a). Our first goal was to evaluate the accumulation of azide-tagged antibiotics across biological contexts related to efflux pumps (Fig. 4b)[55]. Efflux pumps can greatly reduce the concentrations of antibiotics by excreting them to the extracellular space. Of the 36 known or putative efflux pumps in *E. coli*[56], AcrAB-TolC is the best characterized efflux pump and has been described as the most prevalent in decreasing susceptibility to antibiotics[57,58]. Genetic deletion of *tolC* is sufficient to disrupt the tripartite formation of the AcrAB-TolC

complex[59,60]. To test CHAMP in this context, we adopted an antibiotic that has been previously described as a canonical substrate of AcrAB-TolC. Novobiocin belongs to the class of aminocoumarin antibiotics, and it inhibits bacterial DNA gyrase[61]. Given its cytosolic target, novobiocin needs to permeate past the OM in order to reach the target gyrase. Deletion and mutations to *tolC* can dramatically alter the MIC of novobiocin, which has been ascribed to its substrate recognition by this powerful efflux pump[62].

**Fig. 4 | Cytosolic accumulation of azide-tagged antibiotics. a** Chemical structure of the 11 azide-tagged antibiotics. **b** Schematic representation of the three biological contexts that were tested with the modified antibiotics. This includes WT *E. coli*, Δ*tolC E. coli*, and WT *E. coli* treated with the small molecule TolC inhibitor PAβN. **c** CHAMP analysis of *E. coli* with 50 μM of novAZ1 in WT *E. coli* or Δ*tolC E. coli* in the presence or absence of PAβN. **d** Comparison of apparent accumulation of 11 azide-tagged antibiotics in *E. coli* using CHAMP across two biological contexts: WT smooth *E. coli* and WT rough *E. coli*. **e** Comparison of apparent accumulation of 11 azide-tagged antibiotics in *E. coli* using CHAMP across two biological contexts: Δ*tolC E. coli* and WT rough *E. coli*. **f** Comparison of apparent accumulation of 11 azide-tagged antibiotics in *E. coli* using CHAMP across two biological contexts: PAβN co-incubated WT rough *E. coli* and Δ*tolC E. coli*. **g** Comparison of apparent accumulation of 11 azide-tagged antibiotics in *E. coli* using CHAMP across two biological contexts: PMBN co-incubated WT rough *E. coli* and WT rough *E. coli*. For all CHAMP assays, *E. coli* cells carrying the HaloTag-expressing plasmid were induced with IPTG, treated with DBCOcl, then incubated with the azide-tagged test molecules for 1 h in PBS at 37 °C. Data are represented as mean +/– SD ($n = 3$) of technical replicates. *P*-values were determined by a two-tailed *t*-test (* denotes a *p*-value < 0.05, ** < 0.01, ***<0.001, ns = not significant).

An azide-modified novobiocin derivative, novAZ1, that minimally altered the chemical configuration of the original antibiotic (Fig. 4a). In WT *E. coli*, CHAMP analysis revealed that novAZ1 had a low level of apparent accumulation relative to *tolC*-null *E. coli* (Fig. 4c), which is an indication that TolC could reduce the effective concentration of novAZ1 in the cytosol of WT *E. coli*. These results are consistent with novAZ1 being a substrate of TolC. A parallel strategy was also employed to test the viability of CHAMP in measuring substrate recognition in *E. coli*. It is well established that the function of TolC can be disrupted with small molecule inhibitors such as phenylalanine-arginine b-naphthylamide (PAβN)[63]. To this end, a CHAMP analysis was performed for novAZ1 with and without the co-incubation of PAβN. Our results showed that the use of a small molecule chemical inhibitor of TolC led to an elevated level of apparent accumulation of novAZ1. Collectively, these experiments validate novAZ1 as a substrate of AcrAB-TolC in the context of the CHAMP assay. In the future, we will explore how CHAMP can be paired with high-throughput screens to discover next-generation TolC inhibitors (or those of other efflux pumps).

With these results in hand, we set out to widen our library of azide-tagged antibiotics and measure their apparent accumulation in *E. coli* in four different contexts that operate at the OM. The antibiotics covered a wide range of mechanisms of action and cellular targets. The azide groups were added via late-stage modifications, and the chemistry used varied depending on the functional groups available to perform edits to install the azide epitope. In all, 11 azide-tagged antibiotics were included in this subpanel, representing one of the most comprehensive analyses reported to date. Analysis of the accumulation profiles revealed that CHAMP was readily able to discern the relative cytosolic accumulation levels of a wide range of molecules. Of note, there was no detectable accumulation of vancomycin azide (vanAZ1) within the cytosolic space of *E. coli*, consistent with expectations for this organism. Vancomycin exerts its activity by binding to the D-Ala-D-Ala terminus of the lipid II peptidoglycan precursor, which is presented on the outer leaflet of the inner membrane[64]. However, *E. coli*, and, more broadly, Gram-negative bacteria are intrinsically resistant to vancomycin due to the impermeability of their outer membrane, and only become susceptible when this barrier is disrupted[65]. Nonetheless, the CHAMP assay, as employed in this study, provides information that excludes cytoplasmic accumulation of vancomycin but remains agnostic to its potential presence in the periplasm. Several antibiotics act within the periplasmic space of *E. coli* and other Gram-negative bacteria, including β-lactam antibiotics, which target periplasmic penicillin-binding proteins (PBPs) involved in peptidoglycan synthesis, and glycopeptide antibiotics such as vancomycin, which inhibit cell wall cross-linking[66,67]. The current inability of the CHAMP platform to discriminate periplasmic localization represents a limitation of the assay in its present form. Accordingly, vancomycin serves as an effective negative control in our CHAMP assay for assessing the intracellular accumulation of other antibiotics. Other molecules that were smaller in size (e.g., trimethoprim, triAZ1, and sulfonamides, sulAZ1) showed effective accumulation into the cytosol of *E. coli*. Among molecules with high accumulation levels were oxazolidinones. A similar profile was also observed for smooth *E. coli* that has its full set

of *O*-antigen (Fig. 4d), which indicates that the *O*-antigen chains do not appear to significantly hinder the accumulation levels of these specific molecules.

Next, accumulation analysis via CHAMP was performed across the entire series of antibiotics using either the TolC-null *E. coli* strain or the small molecule AcrAB-TolC inhibitor. Interestingly, a number of antibiotic derivatives, including puromycin, showed differential accumulation in *tolC*-deleted strains relative to the parental WT strain (Fig. 4e). For some of these types of structures, the differential accumulation in *tolC*-null strains had yet to be analyzed. Additionally, accumulation profiles for the antibiotic derivatives were also tested with WT cells treated with the small molecule inhibitor, PAβN. As expected, the profile of accumulation across the series of antibiotics was not identical (Fig. 4f). This difference can be attributed to many factors, including the disruption to the OM in the absence of TolC and/or the polypharmacology of most small molecules, including PAβN. Testing a broader range of antibiotics could start to reveal how cells respond to genetic or small-molecule stressors in the context of drug accumulation in *E. coli*. These results demonstrate the potential of utilizing CHAMP to confirm inhibitors of the TolC efflux pump and, most importantly, offer a means to readily compare the apparent accumulation of antibiotic derivatives in strains with genetic alterations that can potentially modulate the accumulation of drugs.

Another aspect of the OM that CHAMP could investigate is how modifying its overall integrity affects the accumulation of modified antibiotics. This includes OM-embedded channels that facilitate passage across the OM and the chemical permeabilization of this critical bilayer. We projected that for molecules whose accumulation was impeded by the OM, there would be an increase in apparent accumulation after the passage was improved. As described before, PMBN was chosen as a chemical permeabilizer of the OM. Critically, a high number of antibiotics had a shifted apparent accumulation in *E. coli* cells upon the co-treatment with PMBN (Fig. 4g). These results clearly demonstrate that the treatment with OM permeabilizers can alter the level of otherwise less permeable molecules into *E. coli*. Alternatively, we tested a genetic strategy to increase the accumulation levels in *E. coli* in a strain with overexpression of the porin FhuA[68]. Our CHAMP results using the antibiotics panel clearly show that, for some antibiotics, there is a shift in their accumulation profile upon hyperporination. Interestingly, the two methods of improving passage across the OM were distinct and driven by their modes of action. Whereas PMBN may have a broader impact, and, theoretically, also impact the cytoplasmic membrane integrity[69], the substrate scope of FhuA is likely to be much narrower because it is a porin. In this work, an open and non-selective variant of FhuA that does not discriminate based on hydrophilicity, designated "pore" was used; the pore is localized to the outer membrane. The substrate size range for this strain is not clear, but *E. coli* K12 BW25113 expressing this variant of FhuA was shown to be sensitive to vancomycin with a 16-fold reduction in MIC relative to the wild type strain[70]. Nonetheless, these analyses provide an extensive description of the changing landscape of accumulation in four biological contexts that can impact passage and retention within bacteria. To the best of our knowledge, this is one of the most comprehensive

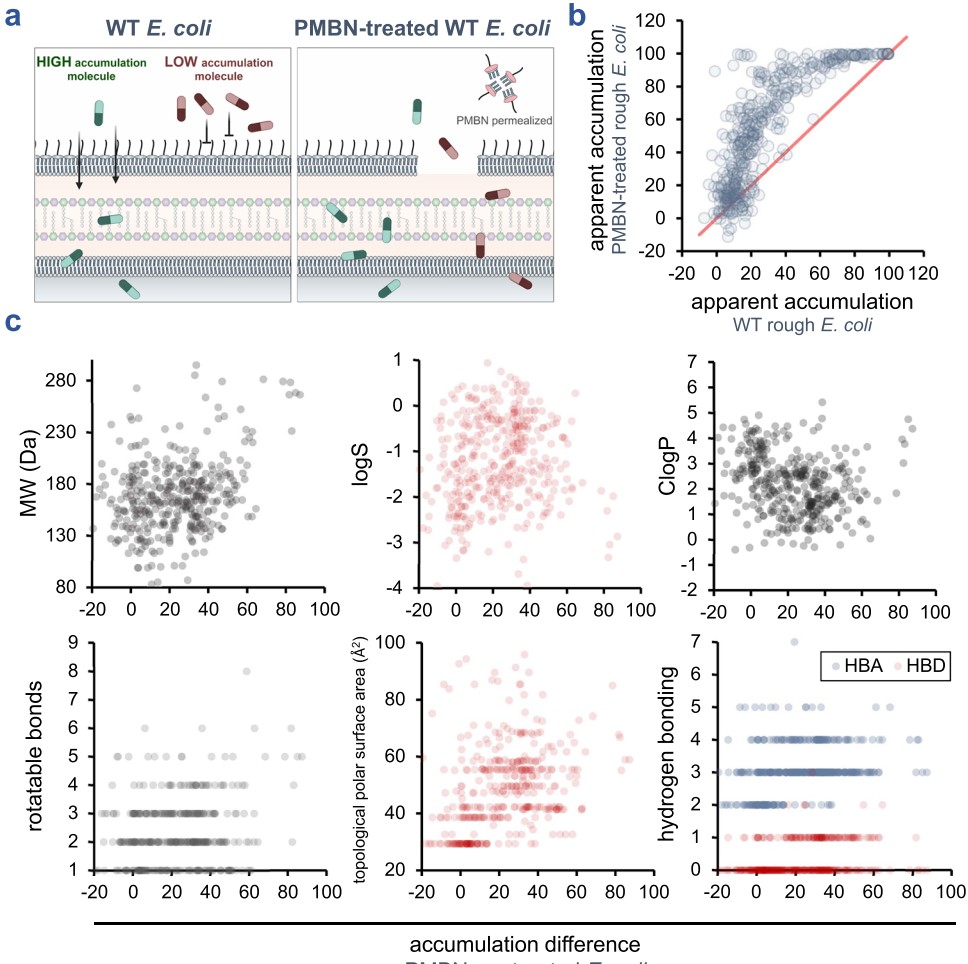

**Fig. 5 | Biological context–dependent differences in cytosolic accumulation.**
**a** Schematic representation of the scenario of a high accumulation molecule and a low accumulation molecule in intact WT *E. coli* relative to WT *E. coli* treated with PMBN. The disruption to the membrane barriers upon PMBN treatment should primarily alter the accumulation profile of molecules whose apparent accumulations are low in intact WT *E. coli*. **b** Comparison of apparent accumulation of 400+ small molecules in *E. coli* using CHAMP across two biological contexts: PMBN co-incubated WT rough *E. coli* and WT rough *E. coli*. **c** Analysis of the 8 different physicochemical parameters that can potentially impact accumulation profiles of the 400+ small molecules (HBA denotes hydrogen bond acceptor; HBD denotes hydrogen bond donor). For all CHAMP assays, *E. coli* cells carrying the HaloTag-expressing plasmid were induced with IPTG, treated with DBCOcl, then incubated with the azide-tagged test molecules for 1 h in PBS at 37 °C.

analyses of a series of antibiotic-derivatives within biological contexts that could impact accumulation.

## High-throughput library analyses in *E. coli*

Our investigation shifted to evaluating larger sets of molecules in *E. coli*. To this end, we assembled a library of 404 commercially available azide-bearing compounds (see SI). This collection encompassed a broad spectrum of characteristics, including hydrophobicity, rigidity, hydrogen bond donors/acceptors, molecular weight, charge, and polarity. Recognizing the prevalence of azide groups in various steric and electronic configurations, we pondered whether these chemical attributes could influence their reactivity with DBCO. To gauge the reaction progress of DBCO-azide within the stipulated assay time-frame, we empirically assessed the reaction completion employing DBCO-modified beads (Fig. S13). A workflow akin to live cell analysis was executed, albeit devoid of the permeability barriers inherent to the cellular outer and inner membranes. Interestingly, it was discerned that not all azides exhibited satisfactory reactivity within the designated timespan of the assay, signifying that azide reactivity necessitates careful consideration (Fig. S14). It is noteworthy that previous investigations have also documented variations in reactivity profiles between azides and strained alkynes, including DBCO[71]. Aromatic

azides have been found to have lower reactivity in SPAAC reactions, with nearly a sevenfold reduced reactivity of phenyl azide in comparison with aliphatic azides, even where the aliphatic azide conformation is in the form of azidotoluene. As for alkyl azides, the following trend in reactivity has been observed: primary > secondary > tertiary. This statement is supported by prior studies[72]. It was quickly recognized that variations in the intrinsic reactivities of azides would constrain the analysis to only those azides exhibiting comparable reactivity, thereby narrowing the chemical diversity and reducing the overall screening breadth achievable with the CHAMP platform.

In light of the challenges posed by the variable reactivities of azides from a large library that are outsourced, we embarked on an alternative approach. To mitigate the influence of azide reactivity, we employed a differential scan, examining the same set of molecules in two closely related biological contexts that exhibit phenotypic divergence. This allowed us to isolate the change in apparent accumulation and attribute it solely to the underlying phenotypic differences. We posited that this method would be apt for unraveling the molecular determinants imposed by the OM of *E. coli*. To illustrate this, our first test case was the analysis of the accumulation of a 404-molecule library in WT *E. coli* with and without the treatment with PMBN (Fig. 5a). As before, the apparent accumulation of small molecules, for which

the OM served as the primary barrier to entry into the cytosol, is expected to change upon co-incubation with PMBN, as this barrier will be disrupted. Conversely, molecules with high propensity for accumulation in intact WT *E. coli* cells would be expected to have minimal differences in their apparent accumulation with PMBN treatment. From our analysis, there was a consistent shift in apparent accumulation across the majority of molecules tested upon the co-incubation with PMBN (Fig. 5b). These results demonstrate the significant barrier that the OM poses to cytosolic arrival and highlight the feasibility of conducting large-scale analyses across various biological contexts. CHAMP can effectively inform us about the differences in the behavior of molecules within these contexts. Further, the difference in the accumulation profile was analyzed across a range of physicochemical parameters, including hydrogen bonding numbers, rotatable bonds, molecular weight, topological polar surface area, ClogP, and logS (Fig. 5c). A larger increase in apparent accumulation following PMBN treatment should theoretically indicate that a specific parameter is implicated in reducing accumulation to the cytosol. These results highlight how high-volume screening can rapidly reveal patterns of molecular features that influence cytosolic accumulation. In this key demonstration, more than 800 individual measurements were made (>400 in wild-type and >400 in PMBN-treated cells), making it the largest reported screen in *E. coli* to date.

As a complement to using a site-selective structural modification series (such as the amino methylation), the late-stage tagging of antibiotics, and the 404 commercial library of azides, we next set out to evaluate the accumulation profile of a larger non-commercial library of small molecules. Each molecule in this library is azide-tagged, making it compatible with CHAMP[31]. These molecules are diverse: aromatic and aliphatic, covering a wide range of chemical space and spanning a range of molecular weights. Our objective was to thoroughly examine the feasibility of a large-scale scan that has not been reported to this point. To further demonstrate the scalability of CHAMP, we screened a library comprising 1152 azide-bearing compounds that was recently described by the Dong and Sharpless groups in their strategy to install azide tags onto molecules using a diazotizing reagent[32]. A benchmarking analysis was performed, spanning a range of cellular fluorescence levels: high, low, and intermediate, using a 96-well plate format. Our experiments affirmed the remarkable reproducibility of CHAMP, underscored by a Z′ score of 0.954 (Fig. S15).

The entire 1152-member library was subjected to bead analysis under the same conditions as the cellular assay. Our results showed that there was a range of reactivities that were observed within the timescale of this assay, owing to the differences in the environments of the azide groups in a large library (Fig. S16). To start, WT *E. coli* cells were subjected to *E. coli* CHAMP analysis, and the results were compared with the reactivity on the beads (Fig. 6a). In this analysis, there is a marked shift in the plot towards the upper left quadrant of the plot. This shift is highly indicative that their engagement with the DBCO landmark is greatly improved upon the removal of the physical barrier (e.g., OM and cytosolic membrane). Subsequently, we applied CHAMP to the complete array of the large library of molecules in four biological contexts: WT *E. coli*, WT *E. coli* pre-incubated with PMBN, hyperporinated *E. coli*, and hyperporinated Δ*tolC E. coli* strains. Our objective was to empirically demonstrate that CHAMP has a throughput capability that is unlike any technology to date, and, when paired with a large library, it provides a workflow to rapidly analyze cytosolic accumulation in *E. coli*. As with the smaller set of molecules, there was a general shift in apparent accumulation in WT *E. coli* upon the pretreatment with PMBN (Fig. 6b). These results demonstrate that chemical disruption to the membrane barriers can broadly improve the cytosolic residency of molecules across a diverse range of physicochemical properties. Hyperporinated *E. coli* could provide a passage point to molecules, as in the case of many antibiotics. In comparison to WT *E. coli*, there was a general increase in apparent accumulation of

molecules in the hyperporinated *E. coli* strain (Fig. 6c). Interestingly, there was general agreement on the two modalities of reduced barriers (Fig. 6d), yet not all compounds behaved the same way. This deviation could be, in part, due to the nature of substrate recognition by porins. Alternatively, the potential for the cytosolic membrane disruption by PMBN could reveal a slightly different set of molecules that arrive in the cytosol. With the goal of leveraging CHAMP to more broadly assess the substrate recognition by TolC, we compared the library response in hyperporinated *E. coli* to hyperporinated Δ*tolC E. coli* (Fig. 6e). This was an attempt to reveal compounds that are TolC substrates but have no access to the sites of recognition due to a lack of accumulation. In this comparison, there was surprisingly general agreement in the behavior of the molecule in the presence and absence of TolC, which could indicate that the reduced barrier in hyperporinated cells could overwhelm the efflux capacity of TolC.

Using the comparison between WT and Δ*tolC* (Fig. 6f), we sought to broadly explore how the structure of the molecule could impact TolC recognition. We examined the correlation between compound efflux and physicochemical properties (Fig. S17). A few properties are shown to correlate with high statistical significance, including hydrogen bond acceptor, conformational flexibility (number of rotatable bonds), amide bond, polar surface area, polarizability, saturated carbon atoms, larger molecular size, and less globularity, which correlate with a higher chance of chemicals being recognized by TolC. There was a large range of chemical structures whose accumulation profiles were found to be especially altered by the deletion of *tolC* (Fig. S18). As part of the effort to understand how much reactivity weighs in on the analysis, we performed correlation analysis with different reactivity cutoff values to filter out compounds with low reactivity. Surprisingly, the reactivity cutoff from 0 to 0.4 does not have a significant influence on correlation values. We theorize that there may be two principal factors at play. First, by performing the differential scan of wild-type and *tolC* deletion mutant, the contribution from reactivity is minimized. Secondly, while the reactivity may still amplify the difference between the wild-type and the Δ*tolC* mutant, the poorer reactivity means a smaller difference in the differential scan. For this reason, such data points make smaller contributions to the final correlation value during the correlation calculation.

Next, we set out to identify motifs that correlate with compound efflux, with the understanding that this would help extract substructure features that make a compound more likely to be recognized by TolC. We initially employed the Bemis-Murcko scaffold method to identify specific scaffolds whose accumulation profiles were particularly sensitive to the presence of TolC. The Bemis-Murcko scaffold is a method that treats rings and the connection between the rings (linkers) as the scaffold. The remaining fragments of the molecule are then considered as side chains. This library of 1152 compounds contains 158 unique scaffolds in total, with some of them much more enriched, e.g., the top 2 scaffolds have more than 10 compounds in each scaffold (Fig. 6g). We selected the first 73 scaffolds having more than 1 member in each scaffold and analyzed the average accumulation shift upon *tolC* deletion. We used a cutoff of 0.05 and depicted scaffolds where the average accumulation shift is more than 0.05. Such scaffolds include piperidine, pyrrolidine, piperazine, thiophene, naphthalene, azetidine, etc. Piperazine-based compounds have been shown to be potential modulators of efflux pump AcrAB-TolC[73,74]. Thiophene antibacterials are known to be substrates of AcrAB-TolC in *E. coli*[75]. Arylpiperidines have been screened as potentiators of novobiocin activity in *E. coli* cells by high-throughput assays[76].

Though Murcko scaffolding represents the skeleton of a molecule, it is possible that side chains may also play an important role in substrate recognition by the TolC-related efflux pumps that are not captured in the scaffold-based analysis. Therefore, we also performed a chemical clustering analysis of the library. Chemical clustering is an approach that finds similarities among the molecules and groups them

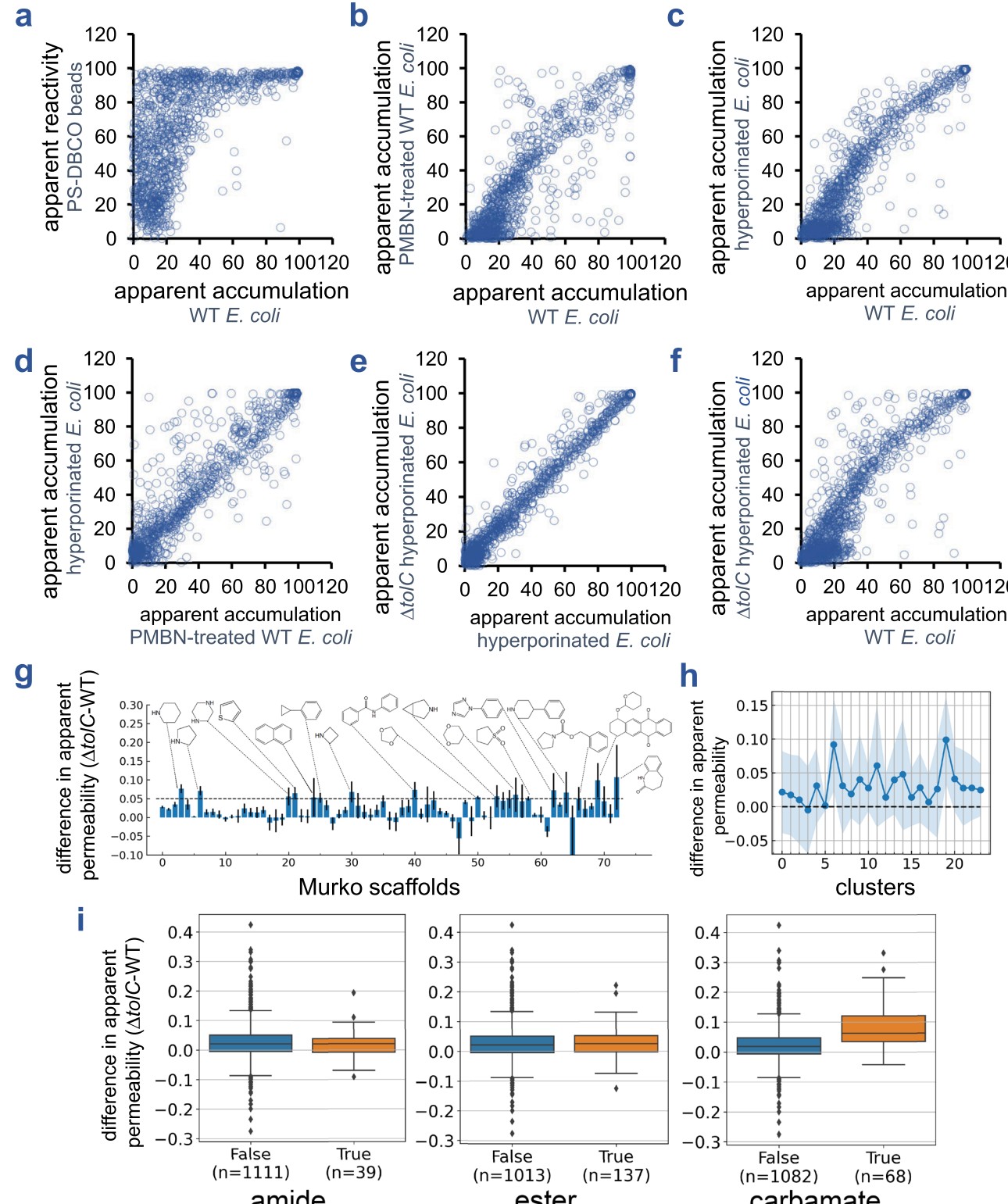

**Fig. 6 | Effects of azide reactivity and physicochemical properties on cytosolic accumulation. a–f** Comparison of apparent accumulation of 1152 small molecules in *E. coli* using CHAMP across two stated biological contexts. **g** Average permeability shift for compounds in each Murcko scaffold. The scaffolds with large permeability shift (>0.05) are annotated with the scaffold chemical structure. **h** Molecular clustering results in 24 clusters, the average permeability shift for each cluster is shown, and shades indicate s.t.d. Clusters 6 and 19 have the lower shaded region also above 0. The representative molecules from these two clusters are annotated below. **i** Permeability shift for compounds with or without (left) amide group, (middle) ester group, (right) carbamate group. Note that the pure carbamate group or the pure ester group here excludes the connected carbamate motif. For all CHAMP assays, *E. coli* cells carrying the HaloTag-expressing plasmid were induced with IPTG, treated with DBCOcl, then incubated with the azide-tagged test molecules for 1 h in PBS at 37 °C.

by similarity. This is also a common approach[77] in drug discovery campaigns to prioritize compounds, select representative molecules, and derive smaller datasets. In our case, we grouped the molecules and investigated whether specific chemical clusters have a larger accumulation shift upon *tolC* deletion. We distributed the library into 24 clusters through hierarchical clustering. We then performed hierarchical clustering. The accumulation shift profile for molecules in each cluster showed two clusters (6 and 19) that have significantly higher apparent accumulation in the absence of TolC (Fig. 6h). Cluster 6 contains many compounds with the carbamate and ester functional group, while cluster 19 contains doxorubicin and a similar derivative. We further developed a hypothesis that the carbamate motif may contribute to compound efflux, though it is not clear if the carbamate or the ester group alone is enough for that. To validate this hypothesis, we calculated accumulation shift upon *tolC* deletion for combinations of these functional groups (Fig. 6i). Only carbamate-based compounds were more populated within this dataset. Together, these results highlight the types of analyses that can be conducted and the queries that can be parameterized using a large volume of accumulation profiles in pathology-relevant biological contexts.

## Discussion

The use of MIC values as a proxy for molecular accumulation has long been a feature of antibiotic drug development. Drug accumulation is necessary but not sufficient for activity, making MICs an imperfect proxy for OM permeability that may generate broadly inaccurate structure-OM accumulation relationships. For example, some of the most widely used antibiotics have complex mechanisms of action that include downstream metabolic dysfunction[78]. As well, structural changes to small molecules affect *both* their OM accumulation and interactions with target molecules. Therefore, it is extremely challenging (and potentially problematic) to attempt to decouple the inherent relationship between accumulation and toxicity solely via MICs. Additionally, anti-virulence agents or adjuvants, such as OM disruptors, typically lack intrinsic antimicrobial activity and are not always suitable for MIC analyses. CHAMP adds a tool to the existing methods used to measure the accumulation of molecules in the cytosol.

It would be beneficial for the community to be able to query beyond the chemical space occupied by existing antibiotics. A wide range of prior efforts have focused on creative ways to measure the accumulation of small molecules, including coupling of periplasmic accumulation to b-lactamase activity[79], split luciferase[14], modification of antibiotics with fluorescent handles[80], LC-MS-based quantification[15,41,42,81,82], electrophysiology using planar lipid bilayers[83,84], and microfluorospectrometry[85–87]. Our laboratories have also recently described a luciferase-based assay using D-cysteine-tagged antibiotics[88]. Despite advances in understanding OM accumulation for certain molecular groups, current approaches have inherent limitations. These limitations preclude the comprehensive analysis of structurally diverse chemical libraries. The lack of this type of analysis poses a significant challenge to establishing guiding principles for structure-accumulation relationships and achieving high predictive power based on chemical structure. Properly elucidating how chemical structure influences accumulation and local concentration – independently of MIC values – is crucial, as these structure-accumulation relationships are fundamental to drug discovery efforts.

Gram-negative bacteria, such as *E. coli*, have an OM and an inner plasma membrane that act as formidable barriers that severely restrict the movement of molecules into and out of the cell. The selective permeability of these membranes can make it challenging for antibiotics to reach intracellular targets. Being able to readily measure intracellular accumulation levels could potentially expedite drug discovery and development. Yet, this type of analysis continues to be a major challenge for the field. In developing CHAMP, we have leveraged the efficient and selective reaction of HaloTag with chloroalkane-modified molecules. Prior efforts using HaloTag in bacteria focused on the tracking of protein fusions[89,90] and the role of cationic peptides on membrane permeabilization[91].

We describe a straightforward and efficient platform, CHAMP, that effectively converts HaloTag expressed in the cytosol of *E. coli* into a strained alkyne. Test molecules tagged with an azide group are imprinted onto HaloTag upon their arrival in the cytosol. As a final step, the level of accumulation is determined by the number of unoccupied sites available for an azide-tagged fluorophore. We demonstrated that fluorescence can be readily measured using flow cytometry or standard plate readers, making it compatible with high-throughput screens. The robustness of the assay was confirmed empirically, and the throughput was exemplified by screening over 1000 individual test molecules. Additionally, we demonstrated that CHAMP can be deployed to report on OM permeabilizers, and we are currently using it to screen for adjuvants that enhance the accumulation levels of antibiotics. Finally, we showed that the high-throughput data from CHAMP can be enhanced with emerging data science and cheminformatics tools to facilitate the discrimination of molecular profiles that increase recognition by the efflux pump AcrAB-TolC. We acknowledge that there are limitations to the CHAMP assay. Principally, in this iteration, the target pathogen must be amenable to genetic manipulation, and the molecule must be tagged with an azide. There is evidence that azides undergo modification within the biological milieu[92–94], a process shown to occur considerably more slowly for alkyl azides than for aryl azides, with cytosolic glutathione contributing to the reduction of azides to amines in *E. coli*. Although the SPAAC DBCO-azide reaction is relatively slow, it proceeds several orders of magnitude faster than the competing glutathione-mediated reduction of azides in biological systems, making this competing process less disruptive to the overall fluorescence readout in our CHAMP assays. Since all azides in the panel are exposed to the same intracellular glutathione environment, the CHAMP assay provides a reliable measure of relative accumulation across different azides under comparable conditions. An additional consideration is the potential impact of impurities in one of the small molecule-azide libraries, which may skew the permeability of the molecules under test. Nevertheless, to minimize any confounding effects from solvent interactions, the concentration of DMSO, known to disrupt membrane integrity, was maintained below 1% in all working solutions, in accordance with the guidelines of the Clinical and Laboratory Standards Institute and previous findings by Tunçer et al.[95].

For antibiotics that do not efficiently cross the OM, an alternative strategy may involve co-treatment with an adjuvant molecule that disrupts the OM, thereby promoting the passage of molecules to their targets. The most widely used OM permeabilizer is PMBN, a fragment of the colistin family of antibiotics. Our results indicate that the OM is a significant barrier to the permeability of a subset of molecules, as its disruption led to a significant increase in fluorescence from azide-tagged dyes. Intriguingly, pre-treatment of cells with PMBN suggests that the disruption of OM integrity in *E. coli* caused by PMBN can persist even after the compound is washed away. Most importantly, we demonstrated that CHAMP could analyze the ability of molecules to permeabilize the OM of *E. coli* by directly measuring the increased permeability of other highly impermeable molecules, such as antibiotics. Given the capacity to adapt the assay to a plate-reading format, we will explore how it can be implemented in future high-throughput screening campaigns for novel OM permeabilizers.

For molecules that navigate past the OM, the next challenge is retention. In drug-resistant bacteria, efflux pumps can reduce the accumulation of molecules that are structurally unrelated[96–98]. While it is generally understood that efflux pumps often play an important role in drug-resistant phenotypes, it remains poorly defined how molecular structure drives recognition by efflux pumps. Assignment of substrates to specific efflux pumps has been made primarily by correlative

**Table 1 | Bacterial strains and plasmids used in this study**

| Strain and Plasmid | Relevant characteristics | Source or reference |
|---|---|---|
| *E. coli* BW25113 | WT K12 Keio parent strain; *E. coli* background | Horizon Discovery Ltd |
| *E. coli* BW25113 *attTn7::mini Tn7T Gmr fhuAC/4L* | K12 Keio parent strain; constitutive expression of the FhuA pore (non-selective variant of FhuA); gentamicin 20 µg/ml) | Brown Lab, McMaster |
| *E. coli* BW25113 *ΔtolC::kanR* | K12 *ΔtolC*, Kan^R (Keio collection, *tolC* deletion strain; kanamycin,50 µg/ml) | Horizon Discovery Ltd |
| *E. coli* BW25113 *ΔtolC::kanR attTn7::mini Tn7T Gmr fhuAC/4L*[70] | K12 *ΔtolC*, Kan^R; constitutive expression of the FhuA pore (non-selective variant of FhuA) Gent^R | Brown Lab, McMaster |
| *E. coli* ATCC25922 | Wild type, smooth, O-antigen | ATCC |
| pQE-30 | *T5* promoter, *Amp*^R | Genscript |

analysis between expression levels and antibiotic susceptibility. Antibiotic susceptibility correlations have many disadvantages that can lead to mis-assignments. The deletion of porins can be a valuable tool to study the recognition of substrates. We showed that CHAMP can be used in combination with the deletion of a primary efflux pump, AcrAB-TolC, to comprehensively test a large library of molecules for their potential recognition. In the future, we plan to start with EKO-35, a strain that has 35 of the efflux pumps deleted out of the genome[99], and then compare this strain to one in which a single efflux pump is introduced. This method will allow for the more precise assignment of structural preference among the various efflux pumps in *E. coli*.

Another major cellular machinery that can modulate the concentration of small molecules is porins embedded within the OM of *E. coli*. Most FDA-approved antibiotics are hydrophilic compounds that are believed to penetrate OMs via porins[100–103]. In *E. coli*, cefoxitin has been purported to primarily enter cells via OmpF and OmpC, whereas ampicillin and ertapenem permeate through OmpC. The importance of porins for small molecule permeation is exemplified by the overexpression of the OM siderophore transporter FhuA, which effectively acts like an open and nonselective porin for the uptake of molecules[68,70]. Hyperporinated *E. coli* becomes highly susceptible to large antibiotics like vancomycin, novobiocin, and erythromycin. For many other molecules, the entry mechanism has not been fully established. The balance between porin-mediated entry and efflux pump-mediated ejection out of the cell will ultimately dictate the effective concentration of molecules inside *E. coli*. Despite its medical significance, it has proven to be extremely challenging to precisely examine the role that each component has on permeation profiles.

## METHODS
### Biological methods
**Plasmid construction for HaloTag expression in *E. coli*.** For expression of the HaloTag protein in WT *E. coli*, (smooth, ATCC25922) and in different variants of *E. coli* K12 (BW25113), HaloTag was cloned onto an expression vector pQE-30 (Genscript) and codon-optimized for expression in *E. coli*; the molecular cloning and DNA synthesis were performed by Genscript (Table 1).

**HaloTag protein amino acid sequence.** MASAEIGTGFPFDPHYVEVLGERMHYVDVGPRDGTPVLFLHGNPTSSYVWRNIIPHVAPTHRCIAPDLIGMGKSDKPDLGYFFDDHVRFMDAFIEALGLEEVVLVIHDWGSALGFHWAKRNPERVKGIAFMEFIRPIPTWDEWPEFARETFQAFRTTDVGRKLIIDQNVFIEGTLPMGVVRPLTEVEMDHYREPFLNPVDREPLWRFPNELPIAGEPANIVALVEEYMDWLHQSPVPKLLFWGTPGVLIPPAEAARLAKSLPNCKAVDIGPGLNLLQEDNPDLIGSEIARWLSTLEISGGGGGSGGGGSGGGGSGS

**HaloTag protein expression and bacterial cell culture.** For expression of the HaloTag protein in WT *E. coli*, (smooth, ATCC25922) and in different variants of *E. coli* K12 (BW25113), the expression plasmid Halo_pQE-30 was transformed into *Escherichia coli* and grown on an LB/agar plate with ampicillin (100 µg/mL) at 37 °C overnight. Colonies

were picked and grown first in 5 mL LB broth with ampicillin at 37 °C overnight. A flask containing 35 mL of LB media was then inoculated with the overnight culture at a ratio of 1:100 in the presence of 100 µg/mL ampicillin and cells were grown at 37 °C for 2 h, or until the optical density at 600 nm reached 0.2 when cultures were induced with 1 mM Isopropyl β-d-1-thiogalactopyranoside (IPTG) at 37 °C for 2 h in a shaker incubator. Cells were then pelleted at 3005 x *g* for 3 min. The pellet was washed three times with 1X PBS pH 7.4, following which cells were resuspended in 35 mL of 1X PBS for the assay. The expression and identity of the protein were confirmed by SDS-PAGE. Uncropped and unprocessed scans of the SDS-PAGE gels can be found in the Source Data master EXCEL file.

**One-step assay with rhodamine chloroalkane (R110cl).** 3 mL of HaloTag-expressing cell culture was transferred to 15 mL conical tubes and pelleted by centrifuging. The cells were washed 3X with 3 mL of 1x PBS and finally re-suspended in 3 mL of 1x PBS. The cells were pipetted into wells in a 96-well plate in triplicate, pelleted down by centrifuging for 3 min at 2700 x *g*, and resuspended using 5 µM R110cl. The cells were incubated for 30 min at 37°C, centrifuged for 3 min, washed 1X with PBS, and fixed with 4% formaldehyde in PBS for 30 min statically at room temperature. The cells were subjected to analysis by flow cytometry on the Attune NxT Acoustic Focusing Cytometer (Invitrogen) by exciting using the blue laser with emission at 525 nm for acquisition.

### CHAMP assay development
**Three-step assay: - DBCOcl incubation, azido-tagged small molecule (pulse), and Flaz dye (chase).** 35 mL of HaloTag expressing fresh cell culture was washed with PBS and transferred to 50 mL conical tubes and pelleted by centrifuging. The cells were re-suspended in 35 mL of a 50 µM solution of DBCOcl in PBS and transferred to an Erlenmeyer flask for incubation at 37 °C for 30 min in a shaker incubator. The DBCOcl imprinted cells were centrifuged to pellet at 3000 x *g* using an HEREAUS Multicentrifuge X1 centrifuge (Thermofisher Scientific), washed 3X with PBS, and resuspended in the same volume of PBS as the original cell suspension. 90 µL of this cell culture was added to each well of a pre-prepared 96-well plate containing 10 µL of the azido-tagged test compound from each 500 µM working solution in PBS. The plate was then covered with plastic film and incubated for 1 h at 37 °C in a shaker incubator. Each azido-tagged test molecule was replicated in three wells. Control experiment wells were either on the same plate or a separate plate in the same shaker incubator for 1 h at 37 °C, and each well contained 10 µL of PBS with 90 µL of DBCO-labeled cell culture (both IPTG-induced and non-induced cells), in triplicate. In the end, the plate was then spun at 2700 x *g* for 3 min using a plate holder centrifuge to pellet the cells, the supernatant was discarded, and the cells were washed 1X with PBS. Next, cells in both the experimental wells and the control wells were fixed with 4% formaldehyde in PBS for 30 min and incubated without shaking at room temperature on the benchtop. Following formaldehyde fixation, the cells were

centrifuged and washed 1X with PBS. In the last chase step, 100 µL of 50 µM Flaz was added to the washed cells in each well, resuspended by pipetting up and down, and incubated at 37 °C for 60 min in a shaker incubator. Finally, the cells in the plate were pelleted by centrifuging in a plate centrifuge, washed 1X with PBS, resuspended in 200 µL of PBS, and subjected to analysis by flow cytometry.

**Two-step assay without fixation prior to chase step: - DBCOcl incubation, azido-tagged dye.** DBCO-labeled cells were prepared as described earlier and resuspended in the same volume of PBS as the original cell suspension. 90 µL of the cell culture was added into each well containing 10 µL of the azido-tagged dye at 500 µM in 96-well plates, covered with plastic film, and incubated for 1 h at 37 °C in a shaker incubator. Each azido-tagged molecule or dye was replicated in three wells. The plate was then spun at 2700 x $g$ for 3 min using a plate holder centrifuge, the supernatant was discarded, and the cells were washed 1X with PBS. Next, cells were fixed with 4% formaldehyde in PBS for 30 min statically at room temperature on the benchtop. Following formaldehyde fixation, the cells were washed 1X with PBS and resuspended in 200 µL of PBS for analysis by flow cytometry.

**Two-step assay with fixation prior to chase step: - DBCOcl incubation, formaldehyde fixation, and azido-tagged dye.** DBCO-labeled cells prepared as described earlier were washed 3X with PBS and resuspended in 4% formaldehyde in PBS and incubated statically at room temperature on the benchtop for 30 min. Following formaldehyde fixation, the cells were washed 1X with PBS and resuspended in the same volume of PBS as was in the original cell suspension. For the azido-tagged dyes screen, 90 µL of the cell culture was added into wells containing 10 µL of the azido-tagged dye at 500 µM in 96-well plates, covered with plastic film, and incubated for 1 h at 37 °C in a shaker incubator. Each azido-tagged dye test was replicated in three wells. The plate was then spun at 2700 x $g$ for 3 min, the supernatant was discarded, and the cells were resuspended in 200 µL of PBS for analysis by flow cytometry.

**Assay with PMBN: - DBCOcl incubation, azido-tagged dye coincubation with PMBN**
DBCO-labeled cells were prepared as described earlier and washed 3X with PBS and resuspended in PBS containing 5 µM PMBN. For the azido-tagged small molecule (three-step) or azido-tagged dyes screen, 90 µL of the cell culture was added into each well containing 10 µL of the azido-tagged small molecule/dye at 500 µM in 96-well plates, covered with plastic film, and incubated for 1 h at 37 °C in a shaker incubator. Each azido-tagged molecule or dye was replicated in three wells. The plate was then spun at 2700 x $g$ for 3 min using a plate holder centrifuge, the supernatant was discarded, and the cells were washed 1X with PBS for fixing, as described earlier, before analysis by flow cytometry (two-step), or for the chase step with an azido-tagged dye (three-step), as described earlier.

**1152 azido-tagged test molecules library screen**
The 1152-molecule azido-tagged library in the form of 96-well mother plates (total 12 plates) was obtained from the Dong Lab at the Institute of Translational Medicine, Zhangjiang Institute for Advanced Study (Shanghai Jiao Tong University) and stored at -20 °C. Each well contained 50 µL of individual test molecules at 5 mM in DMSO. Prior to the assay, the original 96-well plates (denoted as the "mother plates") were allowed to warm up to room temperature and spun down for 1 min at 2700 x $g$, to allow the stock solution to settle at the bottom of each well, using a plate holder centrifuge. 10 µL of the DMSO stock was drawn from each well and dispensed to a new set of plates (denoted as the "daughter plates"). Then 90 µL PBS was added to the daughter plates to make a 100 µL solution of each test molecule at 500 µM working concentration. 10 µL of each test molecule from the

daughter plates was transferred to a new 96-well plate (the "working plates").

DBCO-labeled cells were prepared as described earlier and poured into a reservoir. 90µL of DBCO-labeled cell suspension was drawn from the reservoir and dispensed to the working plates prepared above using a 96-multichannel pipette dispenser (Sequence®). The plates were covered with plastic films and incubated for 1 h at 37 °C in a shaker incubator. The plates were then spun at 2700 x $g$ for 3 min using a plate holder centrifuge. The supernatant was discarded, and the cells were washed 1X with PBS. Next, cells were fixed with 4% formaldehyde in PBS for 30 min statically at room temperature on the benchtop. Following formaldehyde fixation, the cells were washed 1X with PBS. For the final chase step, 100 µL of 50 µM Flaz was added to each well, resuspended, and incubated at 37 °C for 1 h in a shaker incubator. Lastly, the cells were pelleted by centrifuging the 96-well plate in a plate holder centrifuge, washed 1X with PBS, resuspended in 200 µL of PBS, and analyzed by flow cytometry.

**Cell preparation for confocal microscopy**
DBCO-labeled cells prepared as described earlier were washed 3X with PBS and resuspended in the same volume of PBS as was drawn from the original cell suspension. For the azido-tagged small molecules screen, 90 µL of the cell culture was added into wells containing 10 µL of the azido-tagged test compound at 500 µM in 96-well plates, covered with parafilm, and incubated for 1 h at 37 °C in a shaker incubator. Each azido-tagged dye test was replicated in three wells. Control experiment wells were either on the same plate or a separate plate in the same incubator for 1 h at 37 °C, and each well contained 10 µL of PBS with 90 µL of DBCO-labeled cell culture (both IPTG-induced and non-induced cells), in triplicate. The plate was then spun at 2700 x $g$ for 3 min in a plate holder centrifuge, the supernatant was discarded, and the cells were washed 1X with 1X PBS.

Next, cells in both the experimental wells and the control wells were fixed with 4% formaldehyde in 1X PBS for 30 min statically incubated at room temperature on the benchtop. Following formaldehyde fixation, the cells were washed 1X with PBS, and for the chase step, 100 µL of 50 µM Flaz was added into each well, resuspended by pipetting up and down, and incubated at 37 °C for 60 min in a shaker incubator. Lastly, the cells were pelleted by centrifuging the 96-well plate in a plate centrifuge, washed 1X with PBS, and resuspended in 200 µL of PBS. For confocal microscopy, the fixed cells were treated with 5 µg/mL of TAMRA-tagged wheat germ agglutinin (Vector Laboratories, RL-1022) for 30 min at 4 °C. Glass microscope slides were spotted with a 1% agar pad, and 5 µL of the sample was deposited onto the agar. Samples were covered with a glass coverslip and imaged using a Zeiss 880/990 multiphoton Airyscan microscopy system (40× oil-immersion lens) equipped with 488 nm and 550 nm lasers. Images were obtained and analyzed via Zeiss Zen software. We acknowledge the Keck Center for cellular imaging and the usage of the Zeiss 880/980 multiphoton Airyscan microscopy system (PI-AP: NIH-OD025156).

**Confocal fluorescence images of *E. coli* cells expressing HaloTag, labeled with D-Lys(FITC), and treated with coumarin chloroalkane to produce a blue cytosolic signal.** *E. coli* cells harboring the plasmid pQE-30_HaloTag (Amp) were cultured overnight in LB medium. The overnight culture was diluted 1:100 into fresh LB medium supplemented with 500 µM D-Lys(FITC) and incubated at 37 °C with shaking until the optical density at 600 nm ($OD_{600}$) reached approximately 1.2 to allow the incorporation of D-Lys(FITC) in the cell wall. Protein expression was induced with 1 mM isopropyl β-D-1-thiogalactopyranoside (IPTG), and cells were incubated for 1 h to allow expression of the HaloTag protein. Uninduced cultures were maintained as negative controls. Next, cells were washed three times with phosphate-buffered saline (PBS) and incubated with 50 µM

coumarin chloroalkane (Comcl) for 30 min at 37 °C with shaking. Cells were subsequently washed three times with PBS, fixed with 4% (w/v) formaldehyde for 30 min, washed again, and mounted on 1% (w/v) agarose pads for imaging by confocal microscopy. Confocal fluorescence imaging was performed using a Leica STELLARIS 8 Super-Resolution Microscope equipped with a 100× oil-immersion objective. Image acquisition and analysis were carried out using ImageJ software. Excitation was performed using 405 nm (Coumarin) and 488 nm (FITC) laser lines. We acknowledge the Keck Center for cellular imaging (PI-AP: NIH-OD025156).

**Confocal fluorescence images of *E. coli* cells expressing HaloTag, labeled with D-Lys(FITC), and treated with DBCOcl, which was subsequently reacted with Comaz to generate a blue cytosolic signal.** *E. coli* cells harboring the plasmid pQE-30_HaloTag (Amp) were cultured overnight in LB medium. The overnight culture was diluted 1:100 into fresh LB medium supplemented with 500 μM D-Lys (FITC) and incubated at 37 °C with shaking until the optical density at 600 nm (OD$_{600}$) reached approximately 1.2 to allow the incorporation of D-Lys (FITC) in the cell wall. Protein expression was induced with 1 mM isopropyl β-D-1-thiogalactopyranoside (IPTG), and cells were incubated for 1 h to allow expression of the HaloTag protein. Uninduced cultures were maintained as negative controls. Next, cells were washed three times with phosphate-buffered saline (PBS) and incubated with 50 μM DBCOcl for 30 min at 37 °C with shaking. Cells were subsequently washed three times with PBS, fixed with 4% (w/v) formaldehyde for 30 min, washed again, and incubated with 50 μM Comaz for 1 h at 37 °C with shaking. Finally, cells were washed three times and mounted on 1% (w/v) agarose pads for imaging by confocal microscopy. Confocal fluorescence imaging was performed using a Leica STELLARIS 8 Super-Resolution Microscope equipped with a 100× oil-immersion objective. Image acquisition and analysis were carried out using ImageJ software. Excitation was performed using 405 nm (Coumarin) and 488 nm (FITC) laser lines. We acknowledge the Keck Center for cellular imaging (PI-AP: NIH-OD025156).

**Novobiocin azide (novAZ1) accumulation assay in WT *E. coli* BW25113 and Δ*tolC E. coli* BW25113 with or without efflux inhibitor PAβN.** DBCO-labeled cells for the accumulation assay were prepared as described earlier. For each strain of *E. coli*, cells were divided into two equal groups, with one set containing 25 μL of 200μM novAZ1 and 25 μL of PBS as a control, and the other set containing 25 μL of 200 μM novAZ1 and 25 μL of 64 μg/mL PAβN. Next, 50 μL of DBCO-labeled WT *E. coli* cell suspension or Δ*tolC E. coli* suspension was added to the wells in volumes needed to bring the final concentration of novAZ1 and PAβN to 50 μM and 16 μg/m,L respectively. These cells were incubated for 1 h, then pelleted by centrifugation, and the supernatant was removed. Cells were washed 3X in PBS and resuspended for the chase step with R110az as described earlier. Finally, the cells were centrifuged, washed 1X with PBS, fixed with 4% formaldehyde statically at room temperature, and subjected to flow cytometry analysis. A similar method was followed for all azido-tagged antibiotics used in this study.

### Uncropped and unprocessed scans
Full scan gels are available on the Zenodo public repository and can be downloaded with no restrictions at https://doi.org/10.5281/zenodo.17434250.

### Statistics and reproducibility
Experiments involving fluorescence microscopy, SDS–PAGE, and flow cytometry were independently repeated at least three times with similar results. For imaging and SDS–PAGE analyses, technical replicates from three wells were pooled prior to immobilization on agar pads for confocal microscopy or denaturation for gel loading. No statistical methods were used to predetermine sample size; sample sizes are consistent with those reported in prior chemical biology studies. For flow cytometry, a minimum of 10,000 events was collected per sample, a conventional threshold that provides acceptable precision under binomial or Poisson sampling assumptions.

### Reporting summary
Further information on research design is available in the Nature Portfolio Reporting Summary linked to this article.

## Data availability
The datasets generated and/or analyzed during the current study are available within this article and its Supplementary Information and can be downloaded with no restrictions at https://doi.org/10.5281/zenodo.17434250. Source data are provided with this paper.

## Code availability
All raw data supporting the findings of this study are available within the article and its Supplementary Information files. Flow cytometry data is available in a spreadsheet that can be downloaded at https://zenodo.org/records/17434251 and is available to the public with no restrictions. A Jupyter notebook for data analysis and the input data for analysis are included in the "Code" folder. This accession code file is available on the Zenodo public repository and can be used to access, with no restrictions, data generated by bioinformatics author Shasha Feng at https://zenodo.org/records/17887446.

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

## Acknowledgements

This study was supported by the NIH grant 1R01AI178975-01 (M.M.P., W.I, and S.S.), R35GM124893 (M.M.P.), R01AI179080-01 (M.M.P., W.I, and S.S.), and the NSF grant MCB-2111728 (W.I.).

## Author contributions

Conceptualization: M.M.P, W.I., and M.S.S.; Methodology: M.M.P, W.I., M.S.S., and G.M.O.; Investigation: G.M.O., Z.L., and S.F.; Formal Analysis: G.M.O., Z.L., S.F., and M.S.G.; Resources: G.M.O., Z.L., M.D.C., R.D., Y.H., B.E.D., T.G., K.B.S., and J.D.; Writing-Original draft: M.M.P., M.S.S., W.I., and GM.O.; Writing- Review and Editing: M.M.P. and G.M.O.; Visualiza-tion: M.M.P.; Supervision: M.M.P, W.I., and M.S.S.; Project Administration and Funding Acquisition: M.M.P, W.I., and M.S.S.

## Competing interests

The authors declare no competing interests.
