## [Transparent Peer Review file · Nature Communications]

Click-Based Determination of Accumulation of Molecules in *Escherichia coli*

Corresponding Author: Dr Marcos Pires

Version 0:

Reviewer comments:

Reviewer #1

(Remarks to the Author)

Summary of what the authors did:

In this manuscript entitled “Click-Based Determination of Accumulation of Molecules in *Escherichia coli*” the authors developed the Chloroalkane Azide Membrane Penetration (CHAMP) assay that quantifies the accumulation of molecules in the cytoplasm of *Escherichia coli*. Previous studies have estimated accumulation of a molecule in the bacteria through the minimal inhibitory concentration (MIC). However, the MIC reflects the molecule’s ability to inhibit a molecular target, as opposed to its accumulation in the cytoplasm.

CHAMP is a combination of the Peptidoglycan Accessibility Click-Mediated Assessment (PAC-MAN) and the Chloroalkane Penetration Assay (CAPA). Bacteria expressing HaloTag are treated with a chloroalkane-tagged strained alkyne (dibenzocyclooctyne; DBCO), anchoring DBCO onto HaloTag. The molecule of interest, modified with a small azide handle, is incubated with the cells and if it reaches the cytoplasm, it will undergo strain-promoted azide-alkyne cycloaddition (SPAAC) with the DBCO-modified HaloTag. Subsequently, bacteria are chased with an azide-tagged fluorescent molecule to capture unoccupied HaloTag-DBCO molecules. Similar to CAPA the fluorescent reading is inversely proportional to the penetration of the molecule of interest.

The authors present careful optimization of CHAMP in *E. coli*. They show that this assay can be applied to individual molecules with small alterations to their structures to create a structure-accumulation relationship (SAR). Next, they apply CHAMP to natural product antibiotics (such as puromycin and vancomycin) and large libraries of molecules (consisting of 404 and 1152 molecules), highlighting the high-throughput nature of CHAMP.

Through CHAMP the authors showed the ability to enhance the accumulation of certain molecules by increasing permeability of the outer membrane and inhibiting the activity of the efflux pump TolC. Careful structure-accumulation analysis highlighted particular structural features that enhanced accumulation and export by TolC. Taken together, CHAMP could be important in driving future antimicrobial development studies by rapidly “weeding out” poor candidates that are unable to accumulate in the bacterial cytoplasm.

General comments

CHAMP is a powerful assay for measuring the accumulation of small molecules in bacteria that has the potential to become a “guiding force” in future antimicrobial design efforts. Importantly the power of the assay stems from being high-throughput, allowing for a large number of reads to be made quickly.

The structure and use of language in the manuscript has led to a clear story with minimal errors. A very strong feature of this manuscript is the extensive meta-analysis of their data, linking the structural features of their molecular library to the efflux exporter TolC and the accumulation of the molecules of interest in the cytoplasm. Moreover, the data presented here is a strong indicator of the potential of antimicrobial cocktails, coupling TolC inhibitors with antibiotics that otherwise do not accumulate as well in *E. coli*.

It is important to note, however, that CHAMP is a “spin-off” of CAPA applying many of its important features (such as HaloTag, proximity ligation and chasing with a fluorophore) to a bacterial system, as opposed to a mammalian one. Also, CHAMP has few differences compared to their recent Bacterial CAPA (BaCAPA) methodology, essentially switching the chemical handle on the molecule of interest and adding chloroalkane-DBCO on HaloTag in *E. coli*. The authors have done a good job acknowledging this.

In light of this being a thorough and well-written manuscript, only minor additional experiments and corrections have been provided below. If they are addressed in a satisfactory manner, I believe this manuscript would make a great addition to the impactful papers published in Nature Communications.

Point-by-point comments:

Introduction (page 4) – The authors should make a stronger case about how there is a big problem with antimicrobial development at the beginning of the introduction and not wait until the discussion to bring this up.

Figure 1 - Very well made. Easy to understand the premise of this assay.

Figure 2d and (page 9) – For clarity, the authors should show an image of the cells without treatment with R110cl and without IPTG treatment to establish the background. This is important as the gels in 2e and 2i suggest leaky expression.

Figure 2e – Authors should appreciate and acknowledge that there is some “leaky” expression of HaloTag (in the absence of IPTG) as shown in lane 3 of the fluorescent gel in figure 2e. It would be more thorough to have an additional lane in that gel where they transform the cells with the same vector lacking the HaloTag gene (that should show no expression) and carry out R110cl treatment to show that there is no “off-target effect”.

Figure 2h – Similar to 2d, to account for leaky expression it would be useful to show an image of the cells that have not treated with DBCOcl, R110cl or IPTG.

Figure 2j – In the left plot, there is also some indication of leaky expression of HaloTag. This is also reflected in Figure S8b and Figure S9.

Figure 3a and page 14 – On page 14 the authors state that for the test molecule, there is “an EC50 of approximately 3 μ M”. Looking at figure 3a, it looks like the EC50 should be higher, closer to 10 or 11 μ M. Should be double checked.

Figure 3a – Since in CHAMP the signal is inversely proportional to the accumulation, it important to mention, in the text and/or figure legend, how the depicted plot was derived.

Figure 5 – For the last panel, HBA (hydrogen bond acceptor) and HBD (hydrogen bond donor) should be defined in the legend for easy comprehension.

Figure 6 – Excellent analysis.

Figure 6i – In the legend of 6i it says “Permeability shift for compounds with or without (left) carbamate group, (middle) ester group, (right) carbamate”. There seems to be a mismatch between what is shown on the figure and what is described in the legend. The left panel in the figure is labelled as “amide” not carbamate.

Discussion, page 32 – “...are porin imbedded within the OM” should be “...is porins imbedded within the OM”.

Define replicates: In figures where experiments were carried out in replicates (such as figures 2, 3, 4 and many supplementary figures), it is important to state if the replicates were technical or biological in the legend of the corresponding figure.

Reviewer #2

(Remarks to the Author)

This manuscript uses HaloTag as a route to anchor a cyclooctyne inside E. coli. Then entry of azide-linked small molecules can be quantified by their competition for reaction of cyclooctyne with a membrane-permeable azide-fluorophore. Entry is quantified for a huge library of different azides, along with analysis in E. coli bearing different alterations to its outer membrane.

I found this study innovative and the experiments thoroughly take forward the concept.

I thought that the data were strong and showed careful statistical analysis. There is one issue to address there: the microscopy in Fig.2 is not informative without showing alongside equivalent negative control images.

I think it is an excellent example of the use of click chemistry in cells to address a problem of medical relevance. I think it will be of high interest to a wide range of scientists.

The authors do a good job in considering the possible effect of the differential reactivity of azides with DBCO. The authors should comment in the Discussion on other possibilities for read-outs from the assay that do not precisely relate to the entry of the small molecule-azide of interest:

Perhaps-

- 1.the small molecule azide gets modified by the cell to some other compound
- some molecule with different entry/exit properties
- the azide could be reduced, so blocking reaction with DBCO
- 2.the small molecule azide alters thiol levels in the cell which changes the non-specific reactivity of the cyclooctyne
- 3.rapid lytic activity of the small molecule-azide
- 4.Engagement of the small molecule-azide with macromolecular targets hindering DBCO reaction
- 5.Potential impact of azide-bearing impurities in each small molecule-azide stock which may have higher permeability.

This is not a criticism of the approach but every assay has the potential for hits with alternative explanations, so it is good to contemplate such possibilities early on and consider how to account for them.

Minor points:

Even with a magnifying glass it is hard to see what is happening to the HaloTag in cells in Fig. 1d. It would be nice to edit the figure so this part is easier for readers to pick up.

Also, it is not ideal that the fluorescence seems to be accumulating in Fig. 1D on the outside of the cell, which could lead to confusion.

Fig. 2l,m It would help readers to draw Cy5az and Cy3az in their most likely charge state

Fig. 5c Give units for surface area

Fig. 6g: Murcko and not Murko

Fig. 6i legend has carbamate twice

“Interestingly, it was discerned that not all azides exhibited satisfactory reactivity within the designated timespan of the assay, signifying that azide reactivity necessitates careful consideration (Fig. S12). It is noteworthy that previous investigations have also documented variations in reactivity profiles between azides and strained alkynes, including DBCO.⁵⁵”

Nice to have one or two sentences to summarize which kinds of azides react badly.

Reviewer #3

(Remarks to the Author)

Ongwae et al present a new method to study molecule uptake into *E. coli*. The paper is an impressive and extensive piece of work that addresses an important scientific problem (how to quantify molecule uptake) and that is of relevance to a critical societal problem (antibiotic resistance). In principle, I feel positively about this manuscript as I think the authors did a thorough job in developing and testing their assay and I believe that it will be of good use to the scientific community. However, I have noticed a number of inaccuracies and mistakes, as well as some open questions in the paper and I would like to see these corrected/addressed before endorsing publication.

1. The authors several times embellish the limitations of alternative methodologies and in some cases the statements are plainly wrong. The authors' work should stand for itself without putting other approaches down and wrong facts should certainly not be used as arguments for the superiority of their method. Some examples:

Page 4/5: Subcellular localization can be assessed with MS, if proper cell fractionation protocols with sufficient washing steps are followed. The same applies to fluorescent tags and inherent fluorescence, if used spectroscopically. Even more, the latter can be used in microscopy approaches, which, in most cases, easily allows assessment of subcellular localization. Page 9: There simply is no challenge to distinguish membrane from cytosol in bacteria. The difference between cell surface association, membrane localization, and cytosolic localization are glaringly obvious, even with simple wide-field microscopes. Even the outer and inner membrane can be easily distinguished from each other when following appropriate protocols. The authors even show this themselves with abundantly clear cytosolic fluorescence of their probe.

Please check through the manuscript and remove faulty claims.

2. The authors describe the SYTOX green assay wrong (page 11). Firstly, SYTOX dyes are perfectly cell wall-permeable. The cell wall is the peptidoglycan layer, not the whole cell envelope structure built up of inner membrane, peptidoglycan cell wall, and outer membrane. In order to penetrate, SYTOX dyes need both the outer and inner membrane compromised. Thus, they indicate disruption of membrane (not cell wall) integrity. Further, since the inner membrane needs large pores to allow the passage of the dye, which will kill the cells, SYTOX dyes indicate dead cells. So what the authors have proven here is simply that Halo tag expression does not simultaneously create large holes in the inner and outer membrane of *E. coli*. Please rephrase this part to correct this.

A second issue with this assay is that it simply does not support the authors' conclusions. Because SYTOX dyes need both the inner and outer membrane compromised, a negative result cannot exclude that either one of them may be damaged. If the authors truly want to make the claim that the cell envelope as a whole is not damaged, they must assess inner and outer membrane integrity separately. This could, for example, be achieved using NPN for the outer and DiSC(3)5 for the inner membrane.

3. The authors misrepresent/misinterpret vancomycin. On page 18, they say that no intracellular accumulation of vancomycin was observed, which would be consistent with the lack of biological activity against *E. coli*. Vancomycin binds to lipid II, precisely to the D-Ala-D-Ala motif of the pentapeptide chain. This is a cell surface structure that is presented on the outer leaflet of the inner membrane. Naturally, vancomycin would never be found intracellularly, not even in sensitive bacteria. However, vancomycin is perfectly able to bind to lipid II and inhibit *E. coli*, if the outer membrane is compromised. The authors even show this themselves (page 19). The authors further imply (by lumping vancomycin in with puromycin and others) that what they observe is uptake into the intracellular space. As outlined above, that is not the case with vancomycin, which binds to the inner membrane.

Please correct these errors and also generally pay more attention to distinguishing cell and cell envelope structures and defining uptake appropriately (as in uptake into the periplasm or cytoplasm) as this is of crucial relevance to the content of the paper.

4. In connection with that, the authors (wrongly) remark that microscopy would pose difficulties in distinguishing the

membrane from the cytosol, but their method clearly cannot achieve this as outlined above. I would be really curious to see microscopy assays here. For example, vancomycin and puromycin uptake in WT compared to delta tolC would be a good setup to show (i) whether their method could also be well-suited for microscopy (which would be really exciting in my opinion) and (ii) whether it actually can distinguish subcellular localization.

5. Since the Halo tag is expressed in the cytosol, I am wondering how it could detect vancomycin uptake (which sits in the membrane as outlined above). I can imagine four scenarios: (i) vancomycin flips over the membrane to the inner leaflet and is then able to interact, (ii) the Halo tag is somehow also expressed in the periplasm where it would encounter vancomycin, (iii) the azide label alters vancomycin properties enabling cytosolic translocation, or (iv) the experimental conditions do compromise the inner membrane so vancomycin and/or Halo tag can cross this barrier. Each one of these would have rather crucial implications for the usability of the method and I would strongly suggest to address this.

6. Somewhat related to the same question, I would be really curious if periplasmic localization would be possible to see. Would the system work with periplasmic expression?

7. The authors make some mistakes describing PMBN. Firstly, they keep talking about sublethal concentrations (page 12/13), which implies that PMBN is lethal. However, it has no activity against *E. coli* (under standard laboratory conditions) that would show in standardized MIC assays (MICs usually above 512 µg/mL). Secondly, they state that it could interfere with the inner membrane (page 20). However, that is not the case (see for example PMID 36165741). The reference they cite for this does not state this either. Please correct this.

8. Also regarding this, on page 20 where the authors argue with the possible inner membrane effects of PMBN, it would make much more sense to argue that FhuA has a lower substrate range because it is a porin, which is naturally limited in the size and charge of molecules that it lets pass. PMBN induces large-scale destruction of the OM, letting much larger molecules through including massive molecules like nisin.

9. On page 28 the authors argue with using MIC changes as proxy for uptake. While the argument that MICs depend on target interaction is of course true, it is also weak in the context of the paper as the authors argue that structural changes to compounds will affect target interaction. However, that is also the case for the azide linkers used here. The argument could further be broken down to testing the activity of compounds with and without PMBN or in WT and a permeable mutant, which is the same setup the authors are using here and hence not really that different as a concept.

Similarly, on page 30/31 the authors argue for the usability of their method in screening for potentiators that disrupt outer membrane integrity. While I do not doubt that this could be an application, the argument is also weakened by the observation that their assay is itself impacted by the integrity of the outer membrane, making such measurements hard to calibrate.

10. Related to the last point, I would wish that the authors would more honestly and critically reflect on the limitations of their method and in an open and honest way compare the pros and cons between their method and existing ones. The embellishment of the limitations of competing techniques and the absence of honest reflections on the limitations of theirs is a red thread in the paper that I cannot condone.

Reviewer #4

(Remarks to the Author)

Antibiotics must overcome the cell wall and accumulate at the target site. In particular for Gram-negatives this is a real challenge with no simple technique to quantify the uptake. As the authors point out correctly there is a true need for simple methods allowing to quantify accumulation. Mass spectrometry has seen a huge gain in sensitivity allowing larger throughputs. Here the authors suggest a novel technique to quantify the uptake of a particular class of antibiotics. This method has the potential for high throughput. The work is carefully done with a broad range of proofs and tests. I recommend strongly publication.

Below are a few comments:

- The manuscript is difficult to read for outsiders. First, I suggest a more basic scheme about the function bringing the subject to a broader audience.
- What is the sensitivity, how many bacteria are needed? Could this extend to single cells?
- I congratulate the authors for the amount of work and test. This is a very good basis for other works. However, I read the ms several times to catch the bigger picture. Maybe restructuring might render the ms easier to read? (I understand that its not the primary goal to make it easy).
- Please clarify the limits of the approach.

Version 1:

Reviewer comments:

Reviewer #1

(Remarks to the Author)

My concerns and comments have been addressed sufficiently by the authors in the revisions process and I recommend this manuscript for publication.

Reviewer #2

(Remarks to the Author)

All my comments have been well answered and I recommend accepting the paper.

Reviewer #3

(Remarks to the Author)

The authors have written a very careful and well-structured rebuttal, amended or corrected the manuscript where necessary, and brought forward good arguments where reviewer suggestions were not or only partly followed. For my comments, I am satisfied with the authors' edits and replies and I have no further revisions. I recommend acceptance of the manuscript.

Reviewer #4

(Remarks to the Author)

The author replied to the question. In my opinion their approach is highly interesting for the field and I recommend publication as it is.

Reviewer 1

Comment1:

Introduction (**page 4**) – The authors should make a stronger case about how there is a big problem with antimicrobial development at the beginning of the introduction and not wait until the discussion to bring this up.

Response: Thank you so much and sorry for this confusion; the description of the problem of antimicrobial development has been moved back to the introduction – **page 4**.

Comment 2:

Figure 1 - Very well made. Easy to understand the premise of this essay.

Response: Thank you very much for the kind feedback. We're glad to hear that the design and clarity of **Fig.1** effectively conveyed the assay's premise.

Comment 3:

Figure 2d and (**page 9**) – For clarity, the authors should show an image of the cells without treatment with R110cl and without IPTG treatment to establish the background. This is important as the gels in 2e and 2i suggest leaky expression.

Response: We thank the reviewer for this advice. **Fig. 2d** and **Fig. S1** have been updated to include labeling of the cell wall using D-Lys(FITC)-OH (green), thereby delineating the cell envelope (*Angew Chem Int Ed Engl.* 2012 Dec 7;51(50):12519-23. doi: 10.1002/anie.201206749). Consequently, the cytosolic label in this image was switched to coumarin chloroalkane (Comcl), which fluoresces in a distinct spectral window, enabling clear differentiation of the cytosol from the periplasmic space. Additionally, *E. coli* cells not induced with IPTG and treated with Comcl have been included as a negative control to establish the baseline fluorescence background.

Fig. S1. |

Fig. 2d and Fig. S1. | Confocal fluorescence images of *E. coli* cells expressing HaloTag, labeled with D-Lys(FITC) and treated with coumarin chloroalkane to produce a blue cytosolic signal. *E. coli* cells harboring the plasmid pQE-30_HaloTag (Amp) were cultured overnight in LB medium. The overnight culture was diluted 1:100 into fresh LB medium supplemented with 500 μ M D-Lys(FITC)-OH and incubated at 37 °C with shaking until the optical density at 600 nm (OD_{600}) reached approximately 1.2 to allow the incorporation of D-Lys(FITC)-OH in the cell wall. Protein expression was induced with 1 mM isopropyl β -D-1-thiogalactopyranoside (IPTG), and cells were

incubated for 1h to allow expression of the HaloTag protein. Uninduced cultures were maintained as negative controls. Next, cells were washed three times with phosphate-buffered saline (PBS) and incubated with 50 μ M of coumarin chloroalkane (Comcl shown on panel **a**) for 30 min at 37 °C with shaking. Cells were subsequently washed three times with PBS, fixed with 4% (w/v) formaldehyde for 30 min, washed again and mounted on 1% (w/v) agarose pads for imaging by confocal microscopy. Confocal fluorescence imaging was performed using a Leica STELLARIS 8 Super-Resolution Microscope equipped with a 100 \times oil-immersion objective. Image acquisition and analysis were carried out using ImageJ software. Excitation was performed using 405nm (Coumarin) and 488 nm (FITC) laser lines; panel **b** shows the images in different channels. Scale bar: 2 μ m.

Comment 4:

Figure 2e – Authors should appreciate and acknowledge that there is some “leaky” expression of HaloTag (in the absence of IPTG) as shown in lane 3 of the fluorescent gel in figure 2e. It would be more thorough to have an additional lane in that gel where they transform the cells with the same vector lacking the HaloTag gene (that should show no expression) and carry out R110cl treatment to show that there is no “off-target effect”.

Response: We thank the reviewer for this advice. **Fig. 2e** has been revised to include additional lanes representing cells harboring the empty vector, as well as cells cultured in the absence or presence of IPTG and treated with or without R110cl. These additions provide a more comprehensive comparison of HaloTag expression and labeling under the various experimental conditions.

Fig. 2e. | SDS–PAGE analysis of *E. coli* expressing HaloTag under varying induction and labeling conditions. Coomassie-stained and fluorescence-imaged gels showing HaloTag protein expression and labeling efficiency in *E. coli* cells carrying the HaloTag-expressing plasmid. Cells were analyzed with or without IPTG induction and in the presence or absence of R110cl treatment to evaluate protein expression and labeling performance.

Comment 5:

Figure 2h – Similar to 2d, to account for leaky expression it would be useful to show an image of the cells that have not treated with DBCOcl, R110cl or IPTG.

Response: We thank the reviewer for this helpful suggestion. **Fig. 2h** and **Fig. S4** have been updated to include *E. coli* cells that were not induced with IPTG but treated with coumarin azide to establish the baseline fluorescence background. The use of coumarin azide was necessary because D-Lys(FITC)-OH (green) was employed to label the cell wall, thereby delineating the cell envelope (*Angew Chem Int Ed Engl.* 2012 Dec 7;51(50):12519-23. doi: 10.1002/anie.201206749). Consequently, a cytosolic probe emitting in a distinct spectral window was required to clearly distinguish the cytosol from the periplasmic space.

a

b

Fig. S4. |

Fig. 2h. | Confocal fluorescence images of *E. coli* cells expressing HaloTag, labeled with D-Lys(FITC) and treated with DBCOCl, which subsequently reacted with Comaz to generate a blue cytosolic signal. *E. coli* cells harboring the plasmid pQE-30_HaloTag (Amp) were cultured overnight in LB medium. The overnight culture was diluted 1:100 into fresh LB medium supplemented with 500 μ M D-Lys (FITC)-OH and incubated at 37 °C with shaking until the optical density at 600 nm (OD_{600}) reached approximately 1.2 to allow the incorporation of D-Lys (FITC)-OH in the cell wall. Protein expression was induced with 1 mM isopropyl β -D-1-thiogalactopyranoside (IPTG), and cells were incubated for 1h to allow expression of the HaloTag protein. Uninduced cultures were maintained as negative controls. Next, cells were washed three times with phosphate-buffered saline (PBS) and incubated with 50 μ M DBCOCl for 30 min at 37 °C with shaking. Cells were subsequently washed three times with PBS, fixed with 4% (w/v) formaldehyde for 30 min, washed again, and incubated with 50 μ M Comaz (on panel **a**) for 1h at 37 °C with shaking. Finally, cells were washed three times and mounted on 1% (w/v) agarose pads for imaging by confocal microscopy. Confocal fluorescence imaging was performed using a Leica STELLARIS 8 Super-Resolution Microscope equipped with a 100 \times oil-immersion objective. Image acquisition and analysis were carried out using ImageJ software. Excitation was performed using 405nm (Coumarin) and 488 nm (FITC) laser lines; panel **b** shows the images in different channels. Scale bar: 2 μ m.

Comment 6:

Figure 2j – In the left plot, there is also some indication of leaky expression of HaloTag. This is also reflected in **Figure S8b** and **Figure S9**.

Response: We thank the reviewer for taking note of this. We observed a modest degree of leaky expression of the HaloTag protein from this plasmid, which contributes to the baseline fluorescence detected in our measurements. Notably, the absolute magnitude of this background signal is dye dependent. As shown in **Fig. 2j** (left panel), the dye Flaz exhibits relatively high background fluorescence and non-specific binding compared with other dyes. Although Flaz displays limited accumulation within *E. coli* cells, it is likely associated with cell surface components, resulting in measurable background even after

standard 3× washing. Consequently, when fluorescence from uninduced (–IPTG) cells is normalized to that of induced (+IPTG) cells, the low intracellular accumulation of Flaz yields a reduced signal-to-noise ratio, which can exaggerate the apparent effect of leaky expression. In contrast, as shown in the right panel, the dye Comaz exhibits low background and accumulates more efficiently in *E. coli*, resulting in a higher signal-to-noise ratio and minimizing the relative contribution of leaky expression.

Comment 7:

Figure 3a and **page 14** – On **page 14** the authors state that for the test molecule, there is “an EC50 of approximately 3 μM”. Looking at figure 3a, it looks like the EC50 should be higher, closer to 10 or 11 μM. Should be double checked.

Response: We thank the reviewer for noting this error and apologize for the oversight. The screenshot below shows the exact value as calculated by GraphPad Prism, where the EC₅₀ was determined to be 13.65 μM. This correction has been incorporated into the revised manuscript on **page 15**, where the value is now accurately reported as 14 μM.

EC50 shift, X is log(concentration)		
Best-fit values		
LogEC50Control	13.65	13.65
EC50Ratio		
Bottom	0.00279	0.00279
Top	0.9729	0.9729
HillSlope	0.1425	0.1425
EC50Control	4.49225E+13	4.49225E+13
95% CI (profile likelihood)		
LogEC50Control		

Comment 8:

Figure 3a – Since in CHAMP the signal is inversely proportional to the accumulation, it is important to mention, in the text and/or figure legend, how the depicted plot was derived.

Response: We thank the reviewer for noticing this omission and apologize for the oversight. The legend for **Fig. 3a** has been revised to include details on how the plot was derived. It now reads:

“The graph was obtained by plotting (1 – fold change, FC) against concentration, such that a higher (1 – FC) value corresponds to greater molecular accumulation.”

This clarification has been incorporated into the revised figure legend in the manuscript.

Comment 9:

Figure 5 – For the last panel, HBA (hydrogen bond acceptor) and HBD (hydrogen bond acceptor) should be defined in the legend for easy comprehension.

Response: We thank the reviewer for noting this omission and apologize for the oversight. The legend for **Fig. 5** has been updated to include definitions for the abbreviations HBA and HBD, which now read:

“(HBA denotes hydrogen bond acceptor; HBD denotes hydrogen bond donor)

This clarification has been incorporated into the revised manuscript to ensure consistency and clarity in data presentation.

Comment 10:

Figure 6 – Excellent analysis.

Response: We thank the reviewer for the feedback.

Comment 11:

Figure 6i – In the legend of 6i it says “Permeability shift for compounds with or without (left) carbamate group, (middle) ester group, (right) carbamate”. There seems to be a mismatch between what is shown on the figure and what is described in the legend. The left panel in the figure is labelled as “amide” not carbamate.

Response: We thank the reviewer for noting this and apologize for the confusion. The first occurrence of “**carbamate**” in the legend of **Fig. 6i** has been corrected to “**amide**.” This change is now reflected in the revised manuscript.

Comment 12:

Discussion, **page 32** – “...are porin imbedded within the OM” should be “...is porins imbedded within the OM”.

Response: We thank the reviewer for noting this and apologize for the error. The sentence on **page 33** has been corrected to read: “...is porins embedded within the outer membrane (OM).” This correction has been implemented in the revised manuscript.

Comment 13:

Define replicates: In figures where experiments were carried out in replicates (such as **figures 2, 3, 4** and many **supplementary figures**), it is important to state if the replicates were technical or biological in the legend of the corresponding figure.

Response: We thank the reviewer for noting this and apologize for the omission. In all figures where experiments were performed in replicates, the phrase “of technical replicates” has been added to the corresponding figure legends to clarify the nature of the replicates used.

Reviewer #2

Comment 1:

I found this study innovative and the experiments thoroughly take forward the concept. I thought that the data were strong and showed careful statistical analysis. There is one issue to address there: the microscopy in **Fig.2** is not informative without showing alongside equivalent negative control images.

Response: We thank the reviewer for this valuable suggestion. **Fig. 2d** and **Fig. S1** have been updated to include labeling of the cell wall using D-Lys(FITC)-OH (green), thereby delineating the cell envelope (*Angew Chem Int Ed Engl.* 2012 Dec 7;51(50):12519-23. doi: 10.1002/anie.201206749). Consequently, the cytosolic probe in this image was changed to coumarin chloroalkane (Comcl), which emits in a distinct spectral window, allowing clear differentiation of the cytosol from the periplasmic space. In addition, *E. coli* cells that were not induced with IPTG but treated with Comcl have been included as a negative control to establish the baseline fluorescence background.

Similarly, **Fig. 2h** and **Fig. S4** have been updated to include *E. coli* cells that were not induced with IPTG but treated with DBCOcl followed by coumarin azide (Comaz), to establish the baseline fluorescence background. The use of Comaz was necessary because D-Lys(FITC)-OH (green) was employed to label the cell wall, thereby delineating the cell envelope.

Fig. 2d. |

Fig. S1. |

Fig. 2d and Fig. S1. | Confocal fluorescence images of *E. coli* cells expressing HaloTag, labeled with D-Lys(FITC) and treated with coumarin chloroalkane to produce a blue cytosolic signal. *E. coli* cells harboring the plasmid pQE-30_HaloTag (Amp) were cultured overnight in LB medium. The overnight culture was diluted 1:100 into fresh LB medium supplemented with 500 μ M D-Lys(FITC)-OH and incubated at 37 °C with shaking until the optical density at 600 nm (OD_{600}) reached approximately 1.2 to allow the incorporation of D-Lys(FITC)-OH in the cell wall. Protein expression was induced with 1 mM isopropyl β -D-1-thiogalactopyranoside (IPTG), and cells were incubated for 1h to allow expression of the HaloTag protein. Uninduced cultures were maintained as negative controls. Next, cells were washed three times with phosphate-buffered saline (PBS) and incubated with 50 μ M of coumarin chloroalkane (Comcl shown on panel a) for 30 min at 37 °C with shaking. Cells were subsequently washed three times with PBS, fixed with 4% (w/v) formaldehyde for 30 min, washed again and mounted on 1% (w/v) agarose pads for imaging by confocal microscopy. Confocal fluorescence imaging was performed using a Leica STELLARIS 8 Super-Resolution Microscope equipped with a 100 \times oil-immersion objective. Image acquisition and analysis were

carried out using ImageJ software. Excitation was performed using 405nm (Coumarin) and 488 nm (FITC) laser lines; panel **b** shows the images in different channels. Scale bar: 2 μ m.

Fig. 2h. | Confocal fluorescence images of *E. coli* cells expressing HaloTag, labeled with D-Lys(FITC) and treated with DBCOCl, which subsequently reacted with Comaz to generate a blue cytosolic signal. *E. coli* cells harboring the plasmid pQE-30_HaloTag (Amp) were cultured overnight in LB medium. The overnight culture was

diluted 1:100 into fresh LB medium supplemented with 500 μM D-Lys (FITC)-OH and incubated at 37 °C with shaking until the optical density at 600 nm (OD_{600}) reached approximately 1.2 to allow the incorporation of D-Lys (FITC)-OH in the cell wall. Protein expression was induced with 1 mM isopropyl β -D-1-thiogalactopyranoside (IPTG), and cells were incubated for 1h to allow expression of the HaloTag protein. Uninduced cultures were maintained as negative controls. Next, cells were washed three times with phosphate-buffered saline (PBS) and incubated with 50 μM DBCOcl for 30 min at 37 °C with shaking. Cells were subsequently washed three times with PBS, fixed with 4% (w/v) formaldehyde for 30 min, washed again, and incubated with 50 μM Comaz (on panel **a**) for 1h at 37 °C with shaking. Finally, cells were washed three times and mounted on 1% (w/v) agarose pads for imaging by confocal microscopy. Confocal fluorescence imaging was performed using a Leica STELLARIS 8 Super-Resolution Microscope equipped with a 100 \times oil-immersion objective. Image acquisition and analysis were carried out using ImageJ software. Excitation was performed using 405nm (Coumarin) and 488 nm (FITC) laser lines; panel **b** shows the images in different channels. Scale bar: 2 μm .

Comment 2 (Items 1 & 2):

The authors do a good job in considering the possible effect of the differential reactivity of azides with DBCO. The authors should comment in the Discussion on other possibilities for read-outs from the assay that do not precisely relate to the entry of the small molecule-azide of interest:

Perhaps-

1.the small molecule azide gets modified by the cell to some other compound

-some molecule with different entry/exit properties

-the azide could be reduced, so blocking reaction with DBCO

2.the small molecule azide alters thiol levels in the cell which changes the non-specific reactivity of the cyclooctyne

Response: We thank the reviewer for this insightful comment. In response to queries **1** and **2** above, additional sentences have been incorporated into the section on limitations

(page 31 of the revised manuscript). These additional sentences specifically address potential off-target effects and provide clarification regarding their possible impact on the study's interpretations.

“There is evidence that azides undergo modification within the biological milieu, a process shown to occur considerably more slowly for alkyl azides than for aryl azides, with cytosolic glutathione contributing to the reduction of azides to amines in *E. coli*. Although the SPAAC DBCO–azide reaction is relatively slow, it proceeds several orders of magnitude faster than the competing glutathione-mediated reduction of azides in biological systems, making this competing process less disruptive to the overall fluorescence readout in our CHAMP assays.⁹²⁻⁹⁴ While it is likely that some small-molecule azides undergo modification, we currently lack a method to quantify the extent of this effect. Nevertheless, since all azides in the panel are exposed to the same intracellular glutathione environment, the CHAMP assay provides a reliable measure of relative accumulation across different azides under comparable conditions.”

Manuscript reference 92. Handlon AL, Oppenheimer NJ. Thiol reduction of 3'-azidothymidine to 3'-aminothymidine: kinetics and biomedical implications. *Pharm Res.* 1988 May;5(5):297-9. doi: 10.1023/a:1015926720740. PMID: 3244639.

Manuscript reference 93. Zhang C, Dai P, Vinogradov AA, Gates ZP, and Pentelute LB. Site-Selective Cysteine–Cyclooctyne Conjugation. *Angew. Chem. Int. Ed.* 2018, 57, 6459.

Manuscript reference 94. Bertozzi CR, Sletten EM. Bioorthogonal Chemistry: Fishing for Selectivity in a Sea of Functionality. *Chem. Commun.*, 2013,49, 11007-11022\

Comment 2 (Item 3):

The authors should comment in the Discussion on other possibilities for read-outs from the assay that do not precisely relate to the entry of the small molecule-azide of interest:
Perhaps-

3. rapid lytic activity of the small molecule-azide

Response: We thank the reviewer for this insightful question. In our assay, molecules exhibiting lytic activity may indeed undergo self-promoted uptake, resulting in relatively higher intracellular accumulation. Conversely, certain compounds, such as polymyxin B nonapeptide (PMBN), primarily interact with the outer membrane—perturbing its structure and facilitating the uptake of other molecules without themselves entering the cytosol (*Antimicrob. Agents Chemother.* 38, 374–377 (1994) <https://doi.org/10.1128/aac.38.2.374>)

We acknowledge that some small-molecule azides could exhibit self-promoted uptake; however, this was not directly assessed in the present study. A commonly used method to evaluate this behavior is coincubation with a fluorescent small molecule, where enhanced uptake of the fluorophore indicates lytic activity. (Farmer, S., Li, Z. S. & Hancock, R. E. J. *Antimicrob. Chemother.* 29, 27–33 (1992).

Our CHAMP assay is responsive to molecules with lytic activity—a property we have leveraged to demonstrate enhanced uptake of Cy3az and Cy5az upon coincubation with PMBN (**Fig. 2I**). In future work, we aim to further exploit this capability of the CHAMP platform to identify potential drug adjuvants—molecules that enhance uptake without necessarily reaching the intracellular target themselves.

Comment 2 (Item 4):

The authors should comment in the Discussion on other possibilities for read-outs from the assay that do not precisely relate to the entry of the small molecule-azide of interest:

Perhaps-

4. Engagement of the small molecule-azide with macromolecular targets hindering DBCO reaction.

Response: We agree with the reviewer’s observation. As noted by Bertozzi and colleagues, it is indeed possible that some small-molecule azides undergo modification or interact with cellular components prior to reacting with DBCO. This represents an inherent limitation of our assay, and we currently lack a direct method to quantify or correct for this effect. Nevertheless, since all azides are exposed to the same cellular

environment, the CHAMP assay provides a consistent and reliable measure of relative accumulation among azides, thereby allowing valid comparative analysis despite this constraint. (Bertozzi, C. R. & Sletten, E. M. Bioorthogonal Chemistry: Fishing for Selectivity in a Sea of Functionality. *Chem. Commun.* 49, 11007–11022 (2013).

Comment 2 (Item 5):

The authors should comment in the Discussion on other possibilities for read-outs from the assay that do not precisely relate to the entry of the small molecule-azide of interest:

Perhaps-

5. Potential impact of azide-bearing impurities in each small molecule-azide stock which may have higher permeability.

Response: We appreciate the reviewer’s insightful comment. A paragraph addressing compound purity as a limitation has been added to the manuscript on **page 32**. “The commercial compounds used in this study are reported by the manufacturers to have a purity greater than 95%. However, we acknowledge that trace impurities within individual molecules could influence the observed results. Given the large number of small molecules screened in this work, we did not independently verify the purity of each compound. To minimize any confounding effects from solvent interactions, the concentration of DMSO—known to disrupt membrane integrity—was maintained below 1% in all working solutions, in accordance with the guidelines of the Clinical and Laboratory Standards Institute and previous findings by Tunçer *et al.*” (Tunçer *et al.*, *Int. J. Biol. Macromol.*, 2024, 267(Pt 2):131581. doi:10.1016/j.ijbiomac.2024.131581].

Comment 2 (Summary):

This is not a criticism of the approach but every assay has the potential for hits with alternative explanations, so it is good to contemplate such possibilities early on and consider how to account for them.

Response: We thank the reviewer for the feedback.

Minor point 1:

Even with a magnifying glass it is hard to see what is happening to the HaloTag in cells in **Fig. 1d**. It would be nice to edit the figure so this part is easier for readers to pick up.

Response: We thank the reviewer for this helpful observation. We agree that **Fig.1d** was not sufficiently clear in its original form. The figures have now been revised and enhanced for improved clarity and readability, ensuring that the data presentation is more intuitive to the reader.

Minor point 2:

Also, it is not ideal that the fluorescence seems to be accumulating in **Fig. 1D** on the outside of the cell, which could lead to confusion.

Response: We appreciate the reviewer's insightful observation that the previously presented image could give the impression that fluorescence was accumulating outside the cell. To address this, the cartoons in **Fig.1d** (shown above) have been revised to more accurately represent intracellular localization. Specifically, dark green cytosolic fluorescence now denotes low-accumulating molecules. This change improves visual clarity and aligns the schematic with the experimental observations.

Minor point 3:

Fig. 2l,m It would help readers to draw Cy5az and Cy3az in their most likely charge state

Response: We thank the reviewer for this helpful suggestion. **Fig. 2l** and **Fig. 2m** have been revised to include the most probable charge states of Cy5az and Cy3az expected under the assay conditions. These updates improve the chemical accuracy of the illustrations and help readers better interpret the molecular behavior depicted.

Minor point 4:

Fig. 5c Give units for surface area

Response: Thank you for pointing this out, and we apologize for the earlier omission. The y-axis in **Fig. 5c** has been updated to display the units for TPSA in square angstroms (\AA^2).

Minor point 5:

Fig. 6g: Murcko and not Murko

Response: Thank you for bringing this to our attention, and we apologize for the oversight. The x-axis label in **Fig. 6g** has been corrected to read “Murcko scaffolds” as shown below.

Minor points 6:

Fig. 6i legend has carbamate twice

Response: We thank the reviewer for noting this and apologize for the confusion. The first occurrence of “**carbamate**” in the legend of **Fig. 6i** has been corrected to “**amide**.” This change is now reflected in the revised manuscript.

Minor points 7:

“Interestingly, it was discerned that not all azides exhibited satisfactory reactivity within the designated timespan of the assay, signifying that azide reactivity necessitates careful consideration (**Fig. S12**). It is noteworthy that previous investigations have also documented variations in reactivity profiles between azides and strained alkynes, including DBCO.⁵⁵”

Nice to have one or two sentences to summarize which kinds of azides react badly.

Response: We appreciate the reviewer’s insightful comment. To address this, we have added a sentence on **page 23** discussing azide reactivity that reads as follows: “Aromatic azides have been reported to exhibit significantly lower reactivity in strain-promoted azide–alkyne cycloaddition (SPAAC) reactions, with phenyl azide showing nearly a sevenfold reduction in reactivity compared to aliphatic azides, even when the aliphatic azide adopts an azidotoluene conformation. As for alkyl azides, the following trend in reactivity has been observed: primary > secondary > tertiary.” This statement is supported by prior studies (Dommerholt, J. *et al.* Highly accelerated inverse electron-demand cycloaddition of electron-deficient azides with aliphatic cyclooctynes. *Nat. Commun.* **5**, 5378 (2014). <https://doi.org/10.1038/ncomms6378>, and Zimmerman, E.S. *et al.* Production of site-specific antibody–drug conjugates using optimized non-natural amino acids in a cell-free expression system. *Bioconjug. Chem.* **25**(2), 351–361 (2014).

Reviewer #3

Comment 1 (Item 1):

The authors several times embellish the limitations of alternative methodologies and in some cases the statements are plainly wrong. The authors' work should stand for itself without putting other approaches down and wrong facts should certainly not be used as arguments for the superiority of their method. Some examples:

Page 4/5: Subcellular localization can be assessed with MS, if proper cell fractionation protocols with sufficient washing steps are followed. The same applies to fluorescent tags and inherent fluorescence, if used spectroscopically. Even more, the latter can be used in microscopy approaches, which, in most cases, easily allows assessment of subcellular localization.

Response: We thank the reviewer for this thoughtful observation and apologize for the use of overly strong language in describing the challenges associated with assigning molecular localization in *E. coli*. The text has been revised for clarity and tone, and now reads on **page 5**:

“(b) would ambiguously define molecule localization unless additional, careful fractionation methods are included to show periplasmic or cytoplasmic accumulation.”
This revision more accurately reflects the methodological nuances involved without overstating the limitation.

Comment 1 (Item 2):

Page 9: There simply is no challenge to distinguish membrane from cytosol in bacteria.

The difference between cell surface association, membrane localization, and cytosolic localization are glaringly obvious, even with simple wide-field microscopes. Even the outer and inner membrane can be easily distinguished from each other when following appropriate protocols. The authors even show this themselves with abundantly clear cytosolic fluorescence of their probe.

Please check through the manuscript and remove faulty claims.

Response: We appreciate the reviewer’s insightful comment. In response, we have revised the text to remove language referring to the difficulties in distinguishing cytosolic and periplasmic localization and have replaced it with a clear explanation of how the cytosol was delineated. The revised text now reads on **page 9**:

“To confirm that the accumulation measurements in this study reflect cytoplasmic localization of molecules, the cytoplasmic compartment of *E. coli* cells was delineated by labeling the cell wall with D-Lys(FITC)-OH, which emits green fluorescence (*Angew Chem Int Ed Engl.* 2012 Dec 7;51(50):12519-23. doi: 10.1002/anie.201206749). Consequently, the cytosolic probe for confocal imaging was changed to coumarin chloroalkane (Comcl), which fluoresces in a distinct spectral window, thereby enabling clear spatial separation of cytosolic and periplasmic signals. As a control, *E. coli* cells that were not induced with IPTG but treated with Comcl were included to establish the baseline fluorescence background.

This revision provides a more direct and mechanistic explanation of how cytoplasmic localization was established in the assay.

Comment 2 (Item 1):

The authors describe the SYTOX green assay wrong (page 11). Firstly, SYTOX dyes are perfectly cell wall permeable. The cell wall is the peptidoglycan layer, not the whole cell envelope structure built up of inner membrane, peptidoglycan cell wall, and outer membrane. In order to penetrate, SYTOX dyes need both the outer and inner membrane compromised. Thus, they indicate disruption of membrane (not cell wall) integrity. Further, since the inner membrane needs large pores to allow the passage of the dye, which will kill the cells, SYTOX dyes indicate dead cells. So what the authors have proven here is simply that Halo tag expression does not simultaneously create large holes in the inner and outer membrane of *E. coli*. Please rephrase this part to correct this.

Response: Thank you so much for pointing this out and we are sorry for the misrepresentation of the SYTOX Green dye. The SYTOX Green paragraph now appears on **page 12** and has been modified to read: “To verify the viability of the *E. coli* cells after inducing HaloTag expression and subjecting cells to DBCOCl treatment, we conducted a SYTOX Green analysis.⁴⁶ In live prokaryotic cells, SYTOX Green is excluded from cells with intact permeability barriers.⁴⁷ Our results revealed that the expression of HaloTag and incubation of these cells with DBCOCl and its occupancy by HaloTag in *E. coli* did not compromise the integrity of the bacterial cytoplasmic membrane (**Fig. S4**). This finding emphasizes the suitability of utilizing CHAMP for investigating molecular permeation to the cytosol without causing disruption to the cell’s permeability barriers.”

Comment 2 (Item 2):

A second issue with this assay is that it simply does not support the authors' conclusions. Because SYTOX dyes need both the inner and outer membrane compromised, a negative result cannot exclude that either one of them may be damaged. If the authors truly want to make the claim that the cell envelope as a whole is not damaged, they must assess inner and outer membrane integrity separately. This could, for example, be achieved using NPN for the outer and DiSC(3)5 for the inner

membrane.

Response: We thank the reviewer for this valuable suggestion. The manuscript has been revised accordingly to present conclusions derived solely from the SYTOX Green assay. Additionally, in the Supplementary Information (page **S8**), the term “*integrity*” has been replaced with “*viability*”, as reflected in the updated caption: “**Fig. S4.** Cellular viability examination”. We greatly appreciate the reviewer’s recommendation regarding the NPN and DiSC₍₃₎5 dyes as a tool and we plan to incorporate these assays in future studies to evaluate changes in outer and inner membrane integrity.

Comment 3:

3. The authors misrepresent/misinterpret vancomycin. On **page 18**, they say that no intracellular accumulation of vancomycin was observed, which would be consistent with the lack of biological activity against *E. coli*. Vancomycin binds to lipid II, precisely to the D-Ala-D-Ala motif of the pentapeptide chain. This is a cell surface structure that is presented on the outer leaflet of the inner membrane. Naturally, vancomycin would never be found intracellularly, not even in sensitive bacteria. However, vancomycin is perfectly able to bind to lipid II and inhibit *E. coli*, if the outer membrane is compromised. The authors even show this themselves (**page 19**). The authors further imply (by lumping vancomycin in with puromycin and others) that what they observe is uptake into the intracellular space. As outlined above, that is not the case with vancomycin, which binds to the inner membrane.

Please correct these errors and also generally pay more attention to distinguishing cell and cell envelope structures and defining uptake appropriately (as in uptake into the periplasm or cytoplasm) as this is of crucial relevance to the content of the paper.

Response: We thank the reviewer for pointing this out and apologize for the lack of clarity in our discussion of vancomycin. Our laboratory has studied and published extensively on vancomycin and its effects on cell wall biosynthesis, but in attempting to provide a brief remark on vancomycin accumulation, the phrasing inadvertently came across as

inaccurate (<https://doi.org/10.1002/anie.201704851> and <https://doi.org/10.1039/c7sc02721d>). The text has been revised to more accurately reflect the established mechanism of vancomycin activity and its lack of accumulation in *E. coli* due to the outer membrane barrier.

The text has now been edited to read on **page 19**:

“Of note, there was no detectable accumulation of vancomycin azide (vanAZ1) within the cytosolic space of *E. coli*, consistent with expectations for this organism. Vancomycin exerts its activity by binding to the D-Ala-D-Ala terminus of the lipid II peptidoglycan precursor, which is presented on the outer leaflet of the inner membrane (Molinari, H. et al. Structure of vancomycin and a vancomycin/D-Ala-D-Ala complex in solution. *Biochemistry* 1990, 29, 9, 2271–2277 <https://doi.org/10.1021/bi00461a010>) For vancomycin to accumulate in the cytoplasm of *E. coli*, it would necessarily have to traverse the periplasmic space, where it would encounter its target, thereby rendering the cells sensitive to the antibiotic. However, *E. coli*—and, more broadly, Gram-negative bacteria—are intrinsically resistant to vancomycin due to the impermeability of their outer membrane and only become susceptible when this barrier is disrupted (Vaara, M. et al. Susceptibility of gram-negative bacteria to polymyxin B nonapeptide. *Antimicrobial Agents and Chemotherapy*, 01 Jun 1984, 25(6):701-705. <https://doi.org/10.1128/aac.25.6.701>)

Accordingly, vancomycin serves as an effective negative control in our CHAMP assay for assessing the intracellular accumulation of other antibiotics.”

Comment 4:

In connection with that, the authors (wrongly) remark that microscopy would pose difficulties in distinguishing the membrane from the cytosol, but their method clearly cannot achieve this as outlined above. I would be really curious to see microscopy assays here. For example, vancomycin and puromycin uptake in WT compared to Δ tolC would be a good setup to show (i) whether their method os also be well-suitable for microscopy (which would be really exciting in my opinion) and (ii) whether it actually can distinguish subcellular localization.

Response: We thank the reviewer for this valuable suggestion. Although we have not yet performed such an investigation, we believe that the CHAMP assay would be compatible with confocal microscopy and could be used to compare fluorescence accumulation in cells treated with a molecule such as puromycin in both WT *E. coli* and Δ tolC *E. coli*. In such a setup, it is expected that higher intracellular accumulation would yield a dimmer fluorescence signal.

While confocal microscopy provides excellent spatial resolution, it is limited in throughput and quantitative sensitivity when detecting subtle differences between molecules with comparable, though not identical, accumulation profiles. For this reason, our CHAMP assays utilize flow cytometry, which enables high-throughput analysis of approximately 10,000 cells per sample. This approach allows for the simultaneous evaluation of up to 10 different conditions, providing reliable relative accumulation measurements in under five minutes.

Comment 5:

Since the Halo tag is expressed in the cytosol, I am wondering how it could detect vancomycin uptake (which sits in the membrane as outlined above). I can imagine four scenarios: (i) vancomycin flips over the membrane to the inner leaflet and is then able to interact, (ii) the Halo tag is somehow also expressed in the periplasm where it would encounter vancomycin, (iii) the azide label alters vancomycin properties enabling cytosolic translocation, or (iv) the experimental conditions do compromise the inner membrane so vancomycin and/or Halo tag can cross this barrier. Each one of these would have rather crucial implications for the usability of the method and I would strongly suggest to address this.

Response: We appreciate the reviewer's observation regarding the inclusion of vancomycin in our assays. For molecules that do not penetrate the cytosol, such as vancomycin, the CHAMP assay does not provide sufficient information to determine subcellular localization. Vancomycin was included in our study as a negative control to illustrate this limitation.

The pQE30_HaloTag plasmid used in our experiments expresses the HaloTag protein in the cytosol. Consequently, only molecules that reach the cytosol can be detected in the CHAMP assay, as their presence is reflected by a decrease in fluorescence intensity relative to the untreated control. Conversely, if a molecule yields a fluorescence signal comparable to that of the untreated control, it may have localized to the periplasm or remained outside the cell; however, the assay cannot distinguish between these scenarios. Thus, the CHAMP assay specifically reports cytosolic accumulation but remains agnostic to molecular localization outside the cytosol.

Comment 6:

Somewhat related to the same question, I would be really curious if periplasmic localization would be possible to see. Would the system work with periplasmic expression?

Response: We thank the reviewer for this thought-provoking question. The CHAMP assay can indeed be adapted for use with HaloTag expressed in the periplasm by employing appropriate localization signal sequences to direct protein targeting. While this line of investigation is currently ongoing in our laboratory, it was not included in the present study. We anticipate presenting these findings in future publications.

Comment 7:

The authors make some mistakes describing PMBN. Firstly, they keep talking about sublethal concentrations (page 12/13), which implies that PMBN is lethal. However, it has no activity against E. coli (under standard laboratory conditions) that would show in standardized MIC assays (MICs usually above 512 µg/mL). Secondly, they state that it could interfere with the inner membrane (page 20). However, that is not the case (see for example PMID 36165741). The reference they cite for this does not state this either. Please correct this.

Response: We thank the reviewer for pointing out this error. On **page 13**, the term “non-lethal” has been corrected to “low.” Indeed, PMBN does not disrupt the inner membrane at the 10 mM concentration used in our assays. We have confirmed this experimentally using the membrane potential-sensitive dye DiSC₍₃₎₅ as shown below.

Inner membrane integrity assessment using DiSC₍₃₎₅ following PMBN treatment.

E. coli cells induced to express HaloTag protein were treated with 10 μ M PMBN or 0.2% Triton X-100 for 30 min; untreated cells served as a negative control. A decrease in DiSC₍₃₎₅ fluorescence in PMBN-treated cells indicates quenching due to dye intercalation into the membrane, whereas dequenching occurs upon pore formation and loss of membrane potential as the dye redistributes into the cytoplasm. Cellular fluorescence was measured using a plate reader (excitation = 622 nm; emission = 670 nm). Data are presented as mean \pm s.d. (n = 3 independent samples within a single experiment). Statistical significance was determined using a two-tailed *t*-test ($P < 0.05$, * $P < 0.01$, ** $P < 0.001$, *** $P < 0.0001$; ns = not significant).

Comment 8:

Also regarding this, on page 20 where the authors argue with the possible inner membrane effects of PMBN, it would make much more sense to argue that FhuA has a lower substrate range because it is a porin, which is naturally limited in the size and charge of molecules that it lets pass. PMBN induces large-scale destruction of the OM, letting much larger molecules through including massive molecules like nisin.

Response: We thank the reviewer for this comment. FhuA is a siderophore importer that typically transports substrates in the range of 700–1000 Da, including ferricrocin, ferrichrome, albomycin, and rifamycin CGP 4832 (Ferguson, A. D. et al. *Science* 282, 2215–2220 (1998); <https://doi.org/10.1126/science.282.5397.2215>). In this study, we utilized an open, non-selective variant of FhuA—designated “pore”—which does not discriminate based on substrate hydrophilicity and is localized to the outer membrane. Although the exact substrate size range for this variant has not been fully characterized, *E. coli* K-12 BW25114 expressing the pore form of FhuA has previously been shown to display a 16-fold reduction in the minimum inhibitory concentration (MIC) of vancomycin compared to the wild-type strain, indicating increased permeability (Krishnamoorthy, G. et al. *Antimicrob. Agents Chemother.* 60, 7372–7381 (2016); <https://doi.org/10.1128/AAC.01882-16>).

On **page 21**, the Main Text has been revised to read “because it is a porin. In this work, an open and non-selective variant of FhuA that does not discriminate based on hydrophilicity, designated “pore” was used; the pore is localized to the outer membrane. The substrate size range for this strain is not clear but *E. coli* K12 BW25114 expressing this variant of FhuA was shown to be sensitive to vancomycin with a 16-fold reduction in MIC relative to the wild type strain”.

Comment 9:

On page 28 the authors argue with using MIC changes as proxy for uptake. While the argument that MICs depend on target interaction is of course true, it is also weak in the context of the paper as the authors argue that structural changes to compounds will affect target interaction. However, that is also the case for the azide linkers used here. The argument could further be broken down to testing the activity of compounds with and without PMBN or in WT and a permeable mutant, which is the same setup the authors are using here and hence not really that different as a concept.

Response: We thank the reviewer for this insightful comment. We agree that a minimum inhibitory concentration (MIC) assay is a valuable method for assessing the arrival of

molecules at their intracellular drug target sites. For our azido-tagged molecules, conducting MIC measurements in the presence or absence of PMBN, or using WT and FhuA mutant strains, provides an effective way to account for structural modifications in the small-molecule drugs under investigation. In light of this, we have revised the paragraph to read on **page 29**: “CHAMP adds a tool to the existing methods used to measure accumulation of molecules in the cytosol.”

Comment 10:

Similarly, on page 30/31 the authors argue for the usability of their method in screening for potentiators that disrupt outer membrane integrity. While I do not doubt that this could be an application, the argument is also weakened by the observation that their assay is itself impacted by the integrity of the outer membrane, making such measurements hard to calibrate.

Response: We thank the reviewer for pointing out this apparent contradiction. In the CHAMP assay, potentiators of molecular accumulation can indeed be identified using azido-tagged dyes that exhibit low fluorescence signal-to-noise ratios in the absence of a membrane-disrupting agent. A representative example is shown in **Fig. 2I**, where sulfo-Cyanine3 azide and sulfo-Cyanine5 azide do not accumulate in *E. coli* above background levels when incubated with vehicle alone. However, in the presence of a membrane disruptor such as PMBN, a marked increase in fluorescence intensity is observed, indicating enhanced intracellular accumulation. It follows, therefore, that experimental conditions producing a fluorescence profile similar to that observed with PMBN—when sulfo-Cyanine dyes are co-incubated with a test molecule—would suggest that the molecule possesses membrane-disrupting or permeability-enhancing activity.

Comment 11:

Related to the last point, I would wish that the authors would more honestly and critically reflect on the limitations of their method and in an open and honest way compare the pros and cons between their method and existing ones. The embellishment of the limitations of competing techniques and the absence of honest reflections on their

method's limitations is a red thread in the paper that I cannot condone.

Response: We thank the reviewer for a thorough review of our work, and we have edited the original manuscript to highlight limitations inherent in the assay as follows:

Page 20 “ Several antibiotics act within the periplasmic space of *E. coli* and other Gram-negative bacteria, including β -lactam antibiotics, which target periplasmic penicillin-binding proteins (PBPs) involved in peptidoglycan synthesis, and glycopeptide antibiotics such as vancomycin, which inhibit cell wall cross-linking.⁶⁶⁻⁶⁷ The current inability of the CHAMP platform to discriminate periplasmic localization represents a limitation of the assay in its present form.”

Page 23 “Aromatic azides have been found to have lower reactivity in SPAAC reactions with nearly a sevenfold reduced reactivity of phenyl azide in comparison with aliphatic azides even where the aliphatic azide conformation is in the form of azidotoluene. As for alkyl azides, the following trend in reactivity has been observed: primary > secondary > tertiary. This statement is supported by prior studies.⁷² It was quickly recognized that variations in the intrinsic reactivities of azides would constrain the analysis to only those azides exhibiting comparable reactivity, thereby narrowing the chemical diversity and reducing the overall screening breadth achievable with the CHAMP platform.”

Page 31-32 “We acknowledge that there are limitations to the CHAMP assay. Principally, in this iteration, the target pathogen must be amenable to genetic manipulation, and the molecule must be tagged with an azide. There is evidence that azides undergo modification within the biological milieu, a process shown to occur considerably more slowly for alkyl azides than for aryl azides, with cytosolic glutathione contributing to the reduction of azides to amines in *E. coli*. Although the SPAAC DBCO–azide reaction is relatively slow, it proceeds several orders of magnitude faster than the competing glutathione-mediated reduction of azides in biological systems, making this competing process less disruptive to the overall fluorescence readout in our CHAMP assays.^{92,93,94} ”

Page 32 “An additional consideration is the potential impact of impurities in each small molecule-azide library stock which may skew permeability of the molecules under test. Given the extensive library of small molecules screened in this work, we did not independently assess the purity of each compound.”

Reviewer #4

Comment 1:

- The manuscript is difficult to read for outsiders. First, I suggest a more basic scheme about the function bringing the subject to a broader audience.

Response: We thank the reviewer for this helpful suggestion to simplify the presentation of the manuscript. Accordingly, we have restructured the cartoons, graphed data, figure captions, confocal microscopy images, and corresponding main text descriptions to improve clarity and ensure that the information is presented in a more intuitive and reader-friendly manner.

Comment 2:

- What is the sensitivity, how many bacteria are needed? Could this extend to single cells?

Response: We appreciate the reviewer’s comment regarding single-cell analysis. The primary challenge with applying this method to individual cells lies in the practical difficulty of subjecting a single cell to multiple incubation, centrifugation, and washing steps while reliably recovering that same cell for laser interrogation in the flow cytometer. Given that cells are inevitably lost during these preparative steps, we employ flow cytometry analysis of approximately 10,000 events as a representative population to ensure statistical robustness when comparing treatments.

The accompanying graph illustrates that even when smaller subsets of cells are analyzed, the relative differences between HaloTag-expressing cells (IPTG-induced) and non-induced controls remain consistent. In this experiment, cells induced with 1 mM IPTG to

express HaloTag protein were sequentially labeled with DBCOcl (30 min) and R110az (50 μ M, 1 h), followed by flow cytometric analysis using varying event count thresholds. The data represent technical replicates from six independent experiments, with uninduced cells serving as negative controls.

Analysis of mean fluorescence intensity (MFI) using one-way ANOVA, with the 10,000-event dataset as the control, revealed no statistically significant differences between datasets collected from smaller sample sizes. However, examination of fluorescence intensity distributions using violin plots indicated substantial variability among individual events, including outliers with extremely high or low signals. Such outliers, if disproportionately represented in smaller datasets, could lead to misleadingly high or low fluorescence readings. For this reason, we consider smaller event counts to be unreliable for quantitative comparisons and therefore rely on 10,000-event analyses for consistent and representative measurements.

How many cells are needed to convey information on accumulation differences?

To determine the minimum number of cells required to reliably detect differences in molecular accumulation, *E. coli* cells induced to express the HaloTag protein were treated sequentially with DBCO–Cl followed by 50 μ M R110–azide for 1 h. Uninduced cells served as negative controls. Cellular fluorescence was quantified using flow cytometry with excitation at 488 nm and emission at 525 nm. Flow cytometer settings were adjusted to record 10,000, 5,000, 1,000, 500, 100, 10, or 1 events per sample. Fluorescence intensity data were analyzed as the mean \pm SD from six technical replicates for each stopping condition. Results are presented as mean \pm SD from three independent experiments. Statistical analysis was performed using a one-way ANOVA, with the mean fluorescence value from 10,000 events serving as the reference group.

No statistically significant differences were observed across the event groups ($P = 0.767$), indicating that acquisition of fluorescence data from as few as 500 cells provides a reliable estimate of accumulation comparable to that obtained from 10,000 events. Statistical significance was defined as (* $P < 0.05$, ** $P < 0.01$, *** $P < 0.001$, **** $P < 0.0001$, ns = not significant).

How many cells are needed to convey information on accumulation differences?

To determine the minimum number of cells required to reliably capture differences in molecular accumulation, *E. coli* cells induced to express the HaloTag protein were treated with DBCO–Cl, followed by 50 μ M R110–azide for 1 h. Uninduced cells served as negative controls. Cellular fluorescence was quantified by flow cytometry using excitation at 488 nm and emission at 525 nm. The flow cytometer was configured to record 10,000, 5,000, 1,000, 500, 100, 10, or 1 event(s) per acquisition. The resulting fluorescence distributions are presented as a violin plot, with each event represented along the y-axis according to fluorescence intensity. Data are expressed as mean \pm SD from three independent experiments. Statistical significance was assessed by one-way ANOVA, using the 10,000-event dataset as the reference group. No significant differences were observed across event counts ($P = 0.767$), indicating that fluorescence data derived from as few as 500 cells are sufficient to convey reliable information on accumulation differences. Statistical significance was defined as (* $P < 0.05$, ** $P < 0.01$, *** $P < 0.001$, *** $P < 0.0001$, ns = not significant).

Comment 3:

- I congratulate the authors for the amount of work and test. This is a very good basis for other works. However, I read the ms several times to catch the bigger picture. Maybe restructuring might render the ms easier to read? (I understand that its not the primary goal to make it easy).

Response: We thank the reviewer for this helpful suggestion to simplify the presentation of the manuscript. Accordingly, we have restructured the cartoons, graphed data, figure captions, confocal microscopy images, and corresponding main text descriptions to improve clarity and ensure that the information is presented in a more intuitive and reader-friendly manner.

Comment 4:

Please clarify the limits of the approach.

Response: We thank the reviewer for this valuable comment. In response, we have revised the Main Text to explicitly address additional limitations of the CHAMP assay. Among others, we now highlight how differences in azide–DBCO reactivity can affect assay reliability, particularly when comparing the intracellular accumulation of chemically distinct azides. Although this limitation is partially mitigated by testing each molecule in two biological contexts, variability in chemical reactivity remains an important factor to consider.

To clarify these points, we have added the following text in the manuscript as shown below:

Page 20 “Several antibiotics act within the periplasmic space of *E. coli* and other Gram-negative bacteria, including β -lactam antibiotics, which target periplasmic penicillin-binding proteins (PBPs) involved in peptidoglycan synthesis, and glycopeptide antibiotics such as vancomycin, which inhibit cell wall cross-linking.⁶⁶⁻
⁶⁷ The current inability of the CHAMP platform to discriminate periplasmic localization represents a limitation of the assay in its present form.”

Page 23 “Aromatic azides have been found to have lower reactivity in SPAAC reactions with nearly a sevenfold reduced reactivity of phenyl azide in comparison with aliphatic azides even where the aliphatic azide conformation is in the form of azidotoluene. As for alkyl azides, the following trend in reactivity has been observed: primary > secondary > tertiary. This statement is supported by prior studies.⁷² It was quickly recognized that variations in the intrinsic reactivities of azides would constrain the analysis to only those azides exhibiting comparable reactivity, thereby narrowing the chemical diversity and reducing the overall screening breadth achievable with the CHAMP platform.”

Page 31-32 “We acknowledge that there are limitations to the CHAMP assay. Principally, in this iteration, the target pathogen must be amenable to genetic manipulation, and the molecule must be tagged with an azide. There is evidence that azides undergo modification within the biological milieu, a process shown to

occur considerably more slowly for alkyl azides than for aryl azides, with cytosolic glutathione contributing to the reduction of azides to amines in *E. coli*. Although the SPAAC DBCO–azide reaction is relatively slow, it proceeds several orders of magnitude faster than the competing glutathione-mediated reduction of azides in biological systems, making this competing process less disruptive to the overall fluorescence readout in our CHAMP assays.^{92,93,94} ”

Page 32 “An additional consideration is the potential impact of impurities in each small molecule-azide library stock which may skew permeability of the molecules under test. Given the extensive library of small molecules screened in this work, we did not independently assess the purity of each compound.”

We thank the reviewers for their careful evaluation of this manuscript and attention to detail. We feel that the suggested revisions have helped us craft an improved manuscript that will better serve the community.

Sincerely,

Marcos Pires
Professor of Chemistry
Department of Chemistry
University of Virginia
Charlottesville, VA 22904
Email: mpires@virginia.edu